# Mobile air quality monitoring and comparison to fixed monitoring sites for instrument performance assessment

Andrew R. Whitehill[1], Melissa Lunden[2], Brian LaFranchi[2], Surender Kaushik[1], Paul A. Solomon[2,a]

[1]Center for Environmental Measurement and Modeling, Office of Research and Development, United States Environmental Protection Agency, Research Triangle Park, North Carolina, 27711, United States of America
[2]Aclima, Inc., San Leandro, California, 24577, United States of America
[a]formerly at: Office of Research and Development, United States Environmental Protection Agency, Las Vegas, Nevada, 89919, United States of America

*Correspondence to*: Andrew R. Whitehill (whitehill.andrew@epa.gov)

**Abstract.** Air pollution monitoring using mobile ground-based measurement platforms can provide high quality spatiotemporal air pollution information. As mobile air quality monitoring campaigns extend to entire fleets of vehicles and integrate smaller scale air quality sensors, it is important to address the need for assessing these measurements in a scalable manner. We explore collocation-based evaluation of air quality measurements in a mobile platform using fixed regulatory sites as a reference. We compare two approaches – a standard collocation assessment technique where the mobile platform is parked near the fixed regulatory site for a period of time and an expanded approach using measurements while the mobile platform is in motion in the general vicinity of the fixed regulatory site. Based on the availability of fixed reference site data, we focus on three pollutants (ozone, nitrogen dioxide, and nitric oxide) with distinct atmospheric lifetimes and behaviors. We compare measurements from a mobile laboratory with regulatory site measurements in Denver, Colorado, USA and in the San Francisco Bay Area, California, USA. Our one-month Denver dataset includes both parked collocation periods near the fixed regulatory sites as well as general driving patterns around the sites, allowing a direct comparison of the parked and mobile collocation techniques on the same dataset. We show that the mobile collocation approach produces similar performance statistics, including coefficients of determination and mean bias errors, as the standard parked collocation technique. This is particularly true when the comparisons are restricted to specific road types, with residential streets showing the closest agreement and highways showing the largest differences. We extend our analysis to a larger (year-long) dataset in California, where we explore the relationships between the mobile measurements and the fixed reference sites on a larger scale. We show that using a 40-hour running median converges to within ±4 ppbv of the fixed reference site for nitrogen dioxide and ozone and up to about 8 ppbv for nitric oxide. We propose that this agreement can be leveraged to assess instrument performance over time during large-scale mobile monitoring campaigns. We demonstrate an example of how such relationships can be used during large-scale monitoring campaigns using small sensors to identify potential measurement biases.

 **1 Introduction**

Mobile air pollution monitoring can resolve fine-scale spatial variability in air pollutant concentrations, allowing communities to map air quality down to the scale of tens of meters in a reproducible manner (Apte et al., 2017; Van Poppel et al., 2013). Expanding fleet-based mobile monitoring will allow the mapping of much larger spatial regions over longer periods and with more repetition. This will improve the accuracy of land use regression models (Messier et al., 2018; Weissert et al., 2020) and supplement our understanding of air quality issues in environmental justice regions (Chambliss et al., 2021).

One concern with fleet-based mobile monitoring, especially as it expands to lower-cost and lower-powered instrumentation, is maintaining instrument performance (accuracy, precision, and bias) over time in a mobile environment. While the use of fleets facilitates scaling of mobile monitoring to large geographic scales, such as multiple counties in an urban area or multiple cities across a large state, coordinating vehicles and drivers to across these geographies makes route laboratory-based calibrations costly, time consuming, and impractical. Moreover, instrumentation can behave differently in a field environment than during laboratory testing and calibration (Collier-Oxandale et al., 2020), making field calibrations or collocations essential for quantitative measurement applications. Field validation usually involves the collocation of one or more test instruments with reference-grade instruments at a fixed monitoring site, such as a regulatory site (Li et al., 2022). Frequent collocations with fixed reference sites have been identified as an important component of quality assurance for mobile monitoring campaigns (Alas et al., 2019; Solomon et al., 2020). Collocated measurements within ongoing campaigns can also be used to identify potential measurement issues. For example, Alas et al. (2019) provide an example of how the collocation of two AE51 black carbon monitors was used to identify and correct unit-to-unit discrepancies during an ongoing mobile monitoring campaign. Collocated measurements are also important for validation of other emerging measurement technologies, such as lower cost sensors (Bauerová et al., 2020; Castell et al., 2017; Masey et al., 2018).

For mobile monitoring applications, parking a mobile platform next to a fixed reference site approximates the collocation technique for assessing instrument performance. Collocated parking can be incorporated into driving patterns for large-scale mobile monitoring campaigns. However, parking near a fixed reference site ensures comparability only at that specific location and only under the specific atmospheric conditions over which the collocation occurred. As a result, many repeat collocations must be performed, leading to inefficiencies in data collection and an approach that is not easily scalable to larger mobile monitoring campaigns. In addition, natural spatial variability in pollutant concentrations makes the selection of collocation site important (Alas et al., 2019; Solomon et al., 2020), which can often prove restrictive for extended monitoring campaigns. In addition, strong agreement when collocated at a fixed reference site may not translate directly into accuracy and precision in other environments (e.g., Castell et al., 2017; Clements et al., 2017).

Ongoing mobile ("in motion") comparisons with fixed reference sites are more scalable than frequent side-by-side parked collocations and could provide an important tool for ongoing instrument performance assessments during extended mobile monitoring campaigns. If collocation comparisons can be extended out to kilometer scales and spread across multiple fixed reference sites over the course of a single campaign, the amount of data used to evaluate the mobile measurements will

be increased, the dynamic range of pollutant concentrations being measured will be larger, and more time can be dedicated to meeting the mobile monitoring data objectives. In this study, we compare mobile air pollution measurements to fixed reference site measurements from both parked and mobile collocations during the same campaign. The objective is to determine if changes in instrument performance, such as bias, can be identified in mobile collocations to a similar degree as with stationary collocations. We will look at using mobile-versus-fixed-site comparisons as a function of road type and distance between the vehicle and the fixed reference site and compare them to the parked comparisons.

If mobile collocation is able to quantitatively assess bias in mobile instrumentation, it would allow easier detection of instrument drift over time or sudden changes in instrument performance that could indicate a malfunction. This methodology will not serve as a calibration of the mobile platform or replace traditional calibration and quality assurance techniques; rather, it is meant to supplement traditional techniques to allow earlier identification of measurement issues during ongoing mobile monitoring campaigns. We explore the concept of mobile collocations using fast response (1-Hz or 0.5-Hz) laboratory-grade air pollution monitoring instrumentation which is independently calibrated and subject to strict quality assurance. This allows us to explore the impact of spatial variability in pollutant concentrations and operational differences in the mobile-versus-fixed-site comparisons without being limited by instrument accuracy or precision concerns. This will help us to understand the strengths and limitations of these methods and to quantify the magnitude of biases that could be detected using these methods. This work could be expanded in the future to mid-range instrumentation and smaller scale sensors of various pollutants and further developed into a scalable approach for ongoing instrument performance assessment during fleet-based mobile monitoring campaigns where frequent in-situ calibrations of sensors using traditional methods is not feasible.

For this study, we focus on the pollutants ozone ($O_3$), nitrogen dioxide ($NO_2$), and nitric oxide (NO). This decision is largely based on the availability of both mobile and fixed reference site data for the two studies we analyze. $O_3$ and $NO_2$ are Criteria Pollutants with adverse health effects that are measured and regulated by the United States Environmental Protection Agency (USEPA), and all three species are commonly measured at air quality monitoring sites in the United States of America (USA) and other countries. The data collected in this study comes from two mobile monitoring deployments – one in Denver, Colorado, USA in 2014 and one in the San Francisco Bay Area, California, USA in 2019 – 2020. The first study included both parked and mobile collocations and allows a direct comparison of the two techniques, whereas the second study contains a larger dataset that allows for a deeper exploration of the relationship between distance and temporal aggregation scales for optimizing the comparisons. We also demonstrate a real world application of the method to detect drift in a $NO_2$ sensor deployed as part of Aclima's collection fleet. Finally, we discuss the results within the context of spatial heterogeneity of the observed pollutants and the implications for extending the approach to additional pollutants not included in this study.

**2 Instrumentation used in this study**

We analyze measurements from several different vehicles equipped with the Aclima, Inc. mobile laboratory measurement and acquisition platform (Aclima Inc., San Francisco, California, USA). These measurements come from two

separate deployments in different locations and time periods. For the first deployment, three Google Street View cars (gasoline powered Subaru Imprezas) were outfitted with the Aclima platform in Denver, Colorado, USA during the summer of 2014 (Whitehill et al., 2020). The second set of data comes from Aclima's Mobile Calibration Laboratory (AMCL), a gasoline powered Ford Transit van that we deployed in the San Francisco Bay Area of California in 2019 – 2020. Aclima designed the AMCL to support the field calibration of Aclima's sensor-based Mobile Node devices (AMN) for deployment in the Aclima mobile monitoring fleet. In both studies, the mobile platforms were equipped with high resolution (0.5 hz or 1 hz data reporting rate) reference-grade air pollution instrumentation to measure $O_3$, $NO_2$, and NO. Additional measurements, including black carbon (BC), size-fractionated particle number counts (PN), and other species were also measured during these campaigns but are not discussed in this manuscript. For this manuscript, we focus on measurements that had equivalent mobile and fixed reference site measurements for both studies. A critical part of this work is our comparison of parked and mobile collocations during the 2014 Denver study, for which we had one-minute averaged reference site data for $O_3$, $NO_2$, and NO but not the other measured species.

$O_3$ was measured using ultraviolet (UV) absorption with a gas phase (nitric oxide) $O_3$ scrubber for the ozone-free channel (2B Technologies Model 211). This technology reduces some of the volatile organic compound interferences observed in other UV photometric ozone monitors (Long et al., 2021), and has been designated as a Federal Equivalent Method (FEM) by the USEPA (Designation EQOA-0514-215, 40 FR 79, June 18, 2014, p. 34734 – 34735). The 2B Model 211 only reports ozone at 2-second intervals, so ozone was measured and reported as 2-second averages for the Denver study. For the California study, all ozone data was averaged up to 10-second averages during initial data collection, so 10-second averaged data is used for the analysis. $NO_2$ was measured using cavity attenuated phase shift (Teledyne API Model T500U) and represents a "true $NO_2$" measurement (Kebabian et al., 2005). The Teledyne API Model T500U has been designated as a FEM by the USEPA (Designation EQNA-0514-212, 40 FR 79, June 18, 2014, p. 34734 – 34735). NO was measured using $O_3$ chemiluminescence (Ecophysics CLD64), which is a recognized international standard reference method for measuring NO (e.g., EN 14211:2012). NO and $NO_2$ are both measured as 1-second averages in both studies. These instruments have all been evaluated by strict test criteria and are recognized as reference methods by various regulatory agencies; however, the reference method designations do not apply to the applications (mobile monitoring) and timescales (1 second to 1 hour) assessed here. These instruments were chosen for their demonstrated excellent data quality and performance to serve as a mobile reference for calibration purposes. It is important that we assess these methods using reference-grade equipment so we can focus on variability due to spatial heterogeneity instead of instrument issues. If the methods we develop here are successful with reference-grade equipment, they can be used to evaluate the performance of low-cost sensors, which might not meet the same strict regulatory standards at present but can still provide valuable data in smaller, lower-power, and lower-cost form factors (Castell et al., 2017; Clements et al., 2017; Wang et al., 2021).

## 3 Mobile platforms parked at fixed reference sites in Denver (2014)

### 3.1 Methods

We begin the analysis with the comparison of measurements from a parked mobile platform to those at a nearby fixed reference site. For this analysis, we use data from a mobile measurement campaign that occurred in Denver, Colorado, USA during the summer of 2014. Professional drivers drove three identical mobile air pollution monitoring platforms, consisting of specially equipped Google Street View cars, through the Denver, Colorado greater metropolitan region between July 25[th], 2014 and August 14[th], 2014. The project goals were to evaluate the performance of the mobile monitoring platforms and to develop methods for assessing data quality and platform comparability. The three cars drove coordinated routes in a 5 km area around several regulatory monitoring sites in the Denver, Colorado area, as well as driving around larger (10 km) areas to understand the variability of air pollutants at different spatial scales. At several planned periods during the study, the drivers were instructed to park one or more of the cars near one of the fixed regulatory monitoring sites in the region. These parked collocations were integrated into the experimental plan to facilitate the assessment of data quality of mobile platform by comparison to fixed reference site measurements. The parked collocations lasted about 20 minutes each and included comparisons at the Colorado Department of Public Health and Environment (CDPHE) CAMP (39.751184 °N, 104.987625 °W) and La Casa (39.779460 °N, 105.005124 °W) sites. The drivers were instructed to park as close as possible to the site but were required to park in an available public parking space on a public street. These restrictions limited how close each car could get to the reference site for each collocation and resulted in individual parked collocation locations ranging in distance from the regulatory stations, as shown in Figure 1. In addition, the drivers were instructed to park facing into the wind when possible to minimize the influence of self-sampling, which posed additional restrictions. A visual screening did not reveal any suspected self-sampling periods, so no additional attempts were made to identify or remove such periods. Additional details about the monitoring campaign can be found in the Supplemental Information and in Whitehill et al. (2020). The CDPHE reported hourly Federal Reference Method (FRM) and FEM measurements of $O_3$, $NO_2$, and NO, and also reported one-minute time resolution data for our study period as part of the 2014 DISCOVER-AQ field campaign (https://www-air.larc.nasa.gov/missions/discover-aq/discover-aq.html).

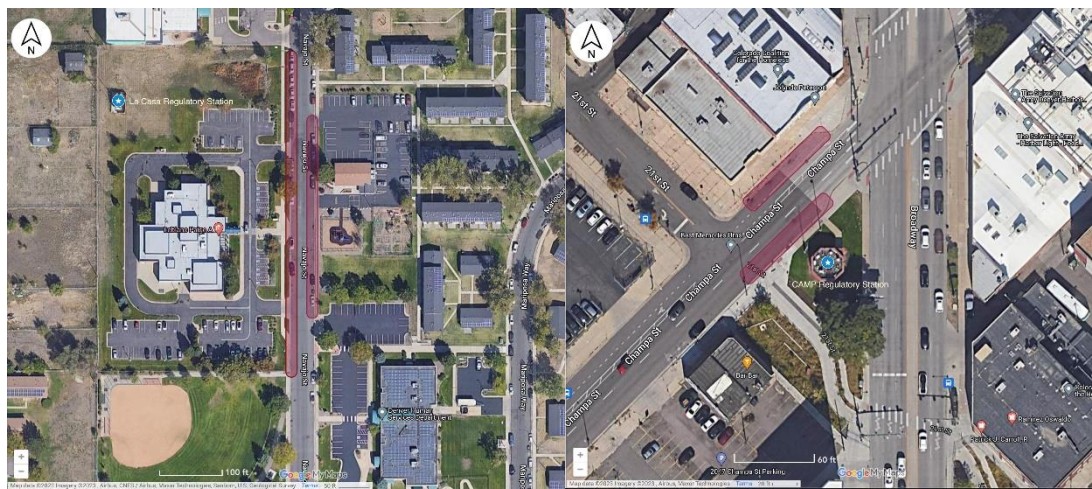

**Figure 1: Satellite view of the area in the immediate vicinity of the La Casa (left) and CAMP (right) regulatory sites. The blue markers denote the regulatory sites and the red shaded areas indicate the range of areas where the vehicles parked during most of the parked collocation periods. Car-to-site distances varied from 80 to 145 m for the La Casa site and from 10 to 85 m for the CAMP site. (Not shown are two periods where the cars parked at the CAMP site just north of the map due to a lack of street parking closer to the site).**

Aclima staff performed quality assurance evaluations on the instruments daily in the field and after the study in the Aclima laboratory. Flow rates remained within instrument specifications throughout the study. We used instrument responses to zero air (from a zero air cylinder) to apply a study-wide zero offset for each instrument on each car. Results from daily span checks for NO (360 ppbv) and $O_3$ (80 ppbv) did not drift beyond the instruments' specifications during the study, so no adjustments were made to instrument span values during the study. All calibrations were performed "through the probe" by connecting a dilution gas calibrator to the sample inlet in a vented tee configuration. A certified NO gas cylinder was diluted by zero air to provide a 360 ppbv NO span gas for the NO instrument, and a certified $O_3$ generator was used to produce 80 ppbv $O_3$ for the $O_3$ span checks. We calibrated the $NO_2$ instrument before and after the study in a laboratory but only performed zero checks on the $NO_2$ instruments in the field.

The bias of the $O_3$ instruments varied between 3% and 6% with a standard deviation of 5%. The bias of the NO instruments varied between 3% and 8% with a standard deviation of 6%. Field bias measurements are based on span checks assuming the linearity of the instrumentation response across the measurement range, which was confirmed in laboratory multi-point calibrations. At the mean observed concentrations during the study (33 ppbv $O_3$ and 53 ppbv NO), the error in span measurements translates to an average bias and precision of 2 ppbv for $O_3$ and an average precision and bias less than 4 ppbv for NO. The NO standard gas had a concentration uncertainty of ±2% (EPA Certified Grade). Mass flow controllers in the dilution calibrator were within their certification period and had a specified accuracy of ±3.6% at the conditions used to generate the span gases. The accuracy of the $O_3$ generator was ±1% and was certified less than three months before the study. We attempted to perform gas-phase titration to produce $NO_2$ span gases in the field, but technical issues prevented us from performing accurate daily span checks on $NO_2$. We did not observe any drift in the $NO_2$ instruments between the pre-campaign

calibration and the post-campaign calibration of the $NO_2$ instruments, so we assumed the calibration of the $NO_2$ instrumentation was constant throughout the campaign period.

At least one of the cars was parked at the CAMP site (within 85 m) for 15 time periods during the study (Table S1) and at the La Casa site (within 150 m) for 16 time periods (Table S2). We previously compared measurements among the three equivalently equipped cars (Whitehill et al., 2020) and determined that one-second $NO_2$ and $O_3$ measurements agreed to within 20% during a day of collocated driving. NO showed higher variability, likely reflecting hyperlocal differences in NO concentrations from exhaust plumes, but were still within 20% about one third of the time. A similar comparison of two Aclima-equipped Google Street View cars in San Francisco and Los Angeles also showed excellent car versus car comparability (Solomon et al., 2020). For the purposes of the present analysis, we assume the data from the three cars is equivalent and interchangeable.

We aggregated all the data up to 1-minute averages (using a mean aggregating function) to put the car measurements on the same timescale as the 1-minute DISCOVER-AQ measurements reported by CDPHE for the CAMP and La Casa sites.

## 3.2 Results and discussion

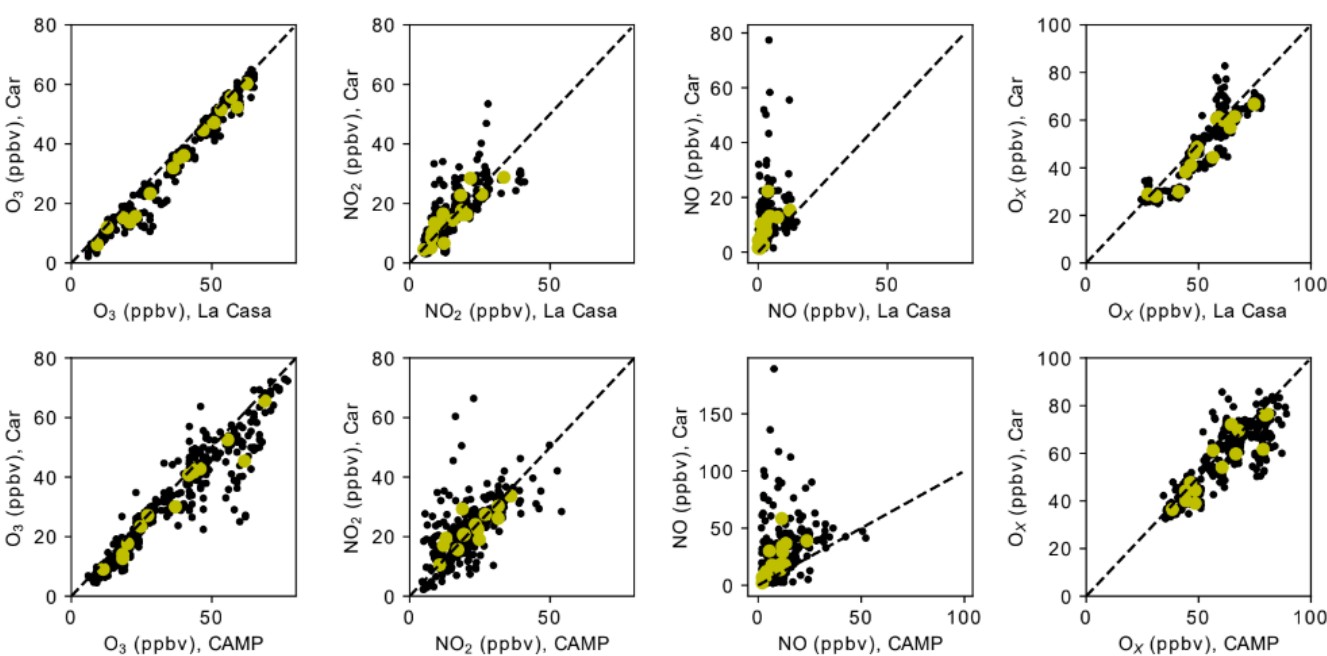

**Figure 2: Scatterplots of 1-minute (black dots) and period mean (yellow dots) car measurements ($O_3$, $NO_2$, NO, and $O_X$) versus fixed reference site (La Casa or CAMP) measurements during the stationary collocation periods. A black dashed 1:1 line is provided for reference.**

Figure 2 shows scatterplots comparing 1-minute $O_3$, $NO_2$, NO, and $O_X$ ($O_3 + NO_2$) for the parked cars versus the fixed reference sites (black circles), as well as the period-specific mean data (yellow circles) and the one-to-one (1:1) line (dashed line). $O_X$ was included in the analysis because it is more likely to be conserved in fresh $NO_X$ emission plumes (assuming most of the $NO_X$ is emitted as NO) than $O_3$ or $NO_2$ separately. We calculated the period-specific means by averaging the discrete 1-minute measurements over the continuous measurement periods that the cars were parked at the fixed reference site (Tables

S1 and S2). We also calculated period-specific medians in a similar way and computed ordinary least squares (OLS) regression statistics for the 1-minute data, the period-specific means, and the period-specific medians (Table S3).

        Figure 2 shows relatively good agreement in the 1-minute observations between parked mobile and stationary reference measurements of $O_3$ and $NO_2$ (and $O_X$), despite some scatter in the relationship. The agreement for NO is poor relative to that for the other pollutants. The coefficients of determination ($r^2$) are highest for $O_3$, then $NO_2$, followed by NO

(see Table S3). With the exception of $O_3$, the linear regression statistics such as slope and intercept did not provide an accurate assessment of bias due to the influence of outlier points on the OLS regression statistics, as evidenced by the relatively low $r^2$ (for $NO_2$ and NO in particular). This relationship is consistent with the expected trends in spatial heterogeneity between the 3 pollutants (Section 7) and illustrates how there can be significant variability in the 1-minute differences between parked mobile and stationary measurements. The $r^2$ does improve somewhat when using the period specific aggregates (mean and median,

Table S3), suggesting that temporal aggregation can reduce some of the variability in the difference and improve the comparisons.

        In order to minimize the impact of scatter in the measurements further, we also looked at the statistics of 1-minute mobile versus fixed site differences, here denoted as $\Delta X = X_{\text{mobile platform}} - X_{\text{reference site}}$. We looked at the mean (i.e., mean bias error), median, standard deviation, and 25th and 75th percentiles of the $\Delta X$ distributions (Table 1). The mean and median of the

$\Delta X$ values effectively aggregates the observations across all of the parked collocation periods and is a more direct assessment of systematic offset bias than the slope and intercept of the OLS regressions (especially for $NO_2$ and NO). This is due to the high sensitivity of the OLS regression statistics to outlier points. Given that the measurements are made under non-ideal conditions, such as these on-the-road, parked (nearby but not spatially coincident) collocations where spatial variability in pollutant concentrations at fine spatial scales appears to be significant, a few extreme outlier points might occur that will skew

the OLS statistics but will have less influence on the $\Delta X$ statistics. For the ease of discussion, in this and subsequent sections we refer to these $\Delta X$ values generally as bias, which includes both spatial and measurement bias.

**Table 1:** Statistics of 1-minute $\Delta X$ comparisons for the stationary collocated periods. Units are ppbv. sd is standard deviation, $P_{25}$ is the 25th percentile, and $P_{75}$ is the 75th percentile.

| | La Casa | | | | | CAMP | | | | |
|---|---|---|---|---|---|---|---|---|---|---|
| | mean | median | sd | $P_{25}$ | $P_{75}$ | mean | median | sd | $P_{25}$ | $P_{75}$ |
| $\Delta O_3$ | -3.6 | -2.8 | 3.3 | -5.3 | -1.2 | -4.3 | -3.4 | 6.9 | -6.5 | -0.5 |

| | | | | | | | | | | |
|---|---|---|---|---|---|---|---|---|---|---|
| $\Delta NO_2$ | 0.0 | -0.9 | 5.5 | -3.0 | 2.6 | 1.0 | 0.5 | 7.6 | -2.8 | 3.8 |
| $\Delta NO$ | 5.1 | 1.9 | 8.8 | 0.9 | 5.9 | 14.2 | 6.8 | 22.0 | 1.6 | 19.7 |
| $\Delta O_X$ | -3.6 | -3.8 | 5.6 | -6.8 | -1.1 | -3.2 | -2.7 | 8.2 | -6.7 | 1.4 |

From Table 1, $NO_2$ shows excellent agreement between the mobile platforms and the fixed reference sites. Mean and median bias values for $NO_2$ were within 1.0 ppbv of 0 for both the CAMP and La Casa sites. $O_3$ shows a minor (but persistent) offset of around 3 ppbv for both sites, which is also apparent from the scatterplots (Figure 2). Both the 25[th] and 75[th] percentiles for $\Delta O_3$ were negative as well, suggesting a real differences in the ozone measurements between the mobile platform and the fixed reference site. The $O_X$ biases were similar to the sum of those for $O_3$ and $NO_2$, as anticipated from our definition of $O_X$. Although the median NO bias for the La Casa site was small (1.9 ppbv), the mean bias for the La Casa site and the mean and median biases from the CAMP site were significantly larger, with values between 5.1 ppbv (for the La Casa mean) and 14.2 ppbv (for the CAMP mean).

The La Casa site is located over 80 m from the nearest street in a predominantly residential neighborhood (Figure 1). The CAMP site, in contrast, is located within meters of the intersection of two major roads (Broadway and Champa St.) and is surrounded by commercial properties. The influence of concentrated direct emission plumes at the mobile platform are a primary source of discrepancies between the mobile platform and the fixed reference site. The relative biases in instrument calibration and differences in concentrations due to the relative proximity of the two instruments to passing emission plumes may also contribute to the discrepancy. From the combined timeseries of all collocations (Figures S4 and S5), short-term peaks in NO and $NO_2$ are present in the mobile platform measurements but not the fixed reference site measurements. This reflects the impact of emission plumes from local traffic. The traffic influences are particularly noticeable at the CAMP site, reflecting its location at a major intersection. While we cannot rule out self sampling of exhaust from the mobile platform, the higher frequency of plume events at the CAMP site versus the La Casa site suggests that local traffic emissions are the primary source of the observed pollution plumes.

The parked collocation results support the assessment of instrument bias; however, the influence of local traffic emissions on the collocation does result in non-optimal conditions. Temporal aggregation can smooth some of the outlier points and make the results more reflective of real measurement differences; however, parametric regression statistics will still be biased by the influence of outlier points. Although it is possible to impose strict collocation criteria for parked collocations that would limit the influence of local emissions, the operational constraints during large-scale mobile monitoring campaigns often necessitate the use of publicly-accessible sites for frequent collocations. Since most scalable parked collocation solutions are likely to be affected by traffic emissions, expanding to allow the use of additional data while driving in the vicinity of the fixed reference station should be explored as a viable alternative. Mobile collocations have the added advantages that spatial biases are averaged out by the motion of the mobile platform through space, effectively allowing each mobile datapoint to sample a larger (and, by extension, more representative) amount of air in the same sampling duration (e.g., Whitehill et al.,

2020). For example, a car traveling 25 meters per second will "sweep" an additional area of 1500 linear meters in one minute compared to the stationary sampling. Thus, regardless of the wind speed, a moving platform will integrate each measurement over a larger area than a stationary platform, making each emission point source have less direct influence on the entire integrated measurement.

## 4 Mobile platforms driving around a fixed reference site in Denver (2014)

### 4.1 Methods

To assess the performance of a mobile collocation approach, we used all the measurements (stationary and moving) collected during the 2014 Denver study, using the parked collocation results as a point of reference. We associated the raw 1-Hz car data with the nearest road using a modified "snapping" procedure (Apte et al., 2017), where we associated each mobile datapoint with the nearest road segment whose direction was within 45° of the car's heading. We assigned each 1-hz datapoint to one of four different road types ("Residential", "Major", "Highway", or "Other") based on the OpenStreetMap (OSM) road classifications of the nearest road segment identified during the snapping procedure (Table S4). We also created an aggregate "Non-Highway" road class, which consisted of roads in the Residential, Major, and Other road classes (i.e., everything not classified as a Highway). Based on our results from Section 3, we believe that the measurements made on Residential roads will generally have lower traffic, and thus stronger agreement with measurements at most fixed reference sites. While travelling on high traffic roads (such as highways), the cars are more likely to be impacted by direct emission plumes. In effect, we are assuming that the OSM road classifications is a general proxy for on-road traffic volume. In addition to road type, we anticipate that the distance between the fixed reference site and the mobile platform will affect the comparisons, with the closest agreement when the distances are small. Although more sophisticated methods are possible to identify and remove high traffic roads, OSM road classifications are a general proxy that can be applied algorithmically over a large portion of the Earth. In contrast, local traffic count data are more sporadic and not always available or easily accessible for the region of interest.

We looked at the mean and median biases and coefficients of determination for measurements made by the cars versus those at the La Casa site. We broke down the analysis by road class and distance buffer, starting at a distance buffer of 500 m and expanding out in 250 m increments. Our goal was to explore how the central tendency of the bias distribution was affected by the cars' distance from the La Casa site, as well as the impact of different road types. We also present (in the Supplemental Information) scatterplots and linear regression statistics for five discrete buffer distances (100 m, 300 m, 1000 m, 3000 m, and 10000 m) and the five different road classes.

### 4.2 Results and discussion

There are fundamental differences between the in-motion collocations discussed in this section and the parked collocations discussed in Section 3. These differences have implications for the interpretations of central tendency (mean or median) bias values and $r^2$ values. In general, the mean and median bias values from both stationary and in-motion collocations

can reflect both measurement bias as well as persistent spatial differences, especially when the instrument inlets are not located at the exact same location. The $r^2$ values generally reflect the variability between the mobile and stationary measurements that results from a combination of measurement precision as well as true spatio-temporal variability. Spatio-temporal variability, in this context, refers to differences in space at the same instant in time, rather than spatial differences that persist over time and are apparent with aggregation over time, which we refer to as systematic spatial bias. At any given instant, the differences between mobile and stationary measurements of different air parcels in different locations are effectively random due to variability in wind direction, wind speed, atmospheric turbulence, and emission rates, among other factors. These factors influence the collection of comparison data points for use in the estimate of $r^2$. For the in-motion observations, both the bias values and the $r^2$ values reflect additional sources of spatial variability that need to be considered as compared to the parked collocations. This is due to the wider range of distances (up to 5 km), varying road types and associated traffic patterns, and potentially different spatial distributions of non-mobile sources in the wider areas covered. As discussed in Section 3, $r^2$ values vary depending on the pollutant measured, and can be quite low even for parked collocations. As a result, we conclude that $r^2$ would not be a good indicator of instrument performance (i.e., precision error) and that using a parametric linear model to attribute gain and offset instrument biases separately is not possible, particularly for NO and $NO_2$. Therefore, the inclusion of $r^2$ in this section is primarily as context for understanding random spatio-temporal variability (as described above) in the comparisons. The focus of this section is to characterize the dimensions over which these spatio-temporal differences, both random and systematic, manifest in the data. By doing so, we hypothesize that we will be able to isolate the conditions under which the variability in bias values can be expected to reflect variability in measurement bias between the mobile and stationary monitors. In particular, we are looking for an optimal operational method for mobile collocation that provides comparable results to that determined from the parked collocations.

The mean and median bias values and $r^2$ values for the car versus the La Casa comparisons are shown as a function of buffer distance and road type in Figure 3. Note here that for each distance D we include all datapoints within a distance of D from the stationary site, so we are not explicitly showing how bias varies with distance from the stationary monitors. We also display the results from the stationary collocation analysis (Section 3) to demonstrate the similarity between the mean and median biases and $r^2$ values from the stationary collocations alongside those from our expanded analysis in this section. Because measurement bias is not expected to correlate with the spatial dimensions featured in these analyses, the difference between road type and with varying distances from the site can be interpreted purely as persistent spatial differences. As shown in Figure 3, measurements from the Highway road class resulted in a significantly higher magnitude of bias compared to other road types, indicating a larger influence of direct emission plumes increasing the variability in concentrations measured on Highways (with respect to the stationary site measurements) than for other road types. The $r^2$ values on Highways are generally lower than on other road types, indicating that there is higher random variability on Highways compared to the stationary site measurements. Differences in bias and $r^2$ between Major and Residential road types are significant in some cases and less so in others, depending upon the pollutant and the distance. However, even in cases where the differences are significant (e.g., $NO_2$ at distances greater than 1000 m), the magnitude of the bias for Major roads is only a fraction of that from Highways. As

discussed above, the degree of bias and random spatial variability between the mobile measurements and the fixed reference sites are primarily impacted by the direct emission plumes on the roadway. We anticipate that Highways have higher traffic and a higher fraction of more heavily polluting vehicles (e.g., heavy duty trucks), which explains the larger magnitude of bias and lower $r^2$ values. Roads with lower expected traffic volumes and less heavy duty vehicles, such as Residential roads, have a lower magnitude of bias.

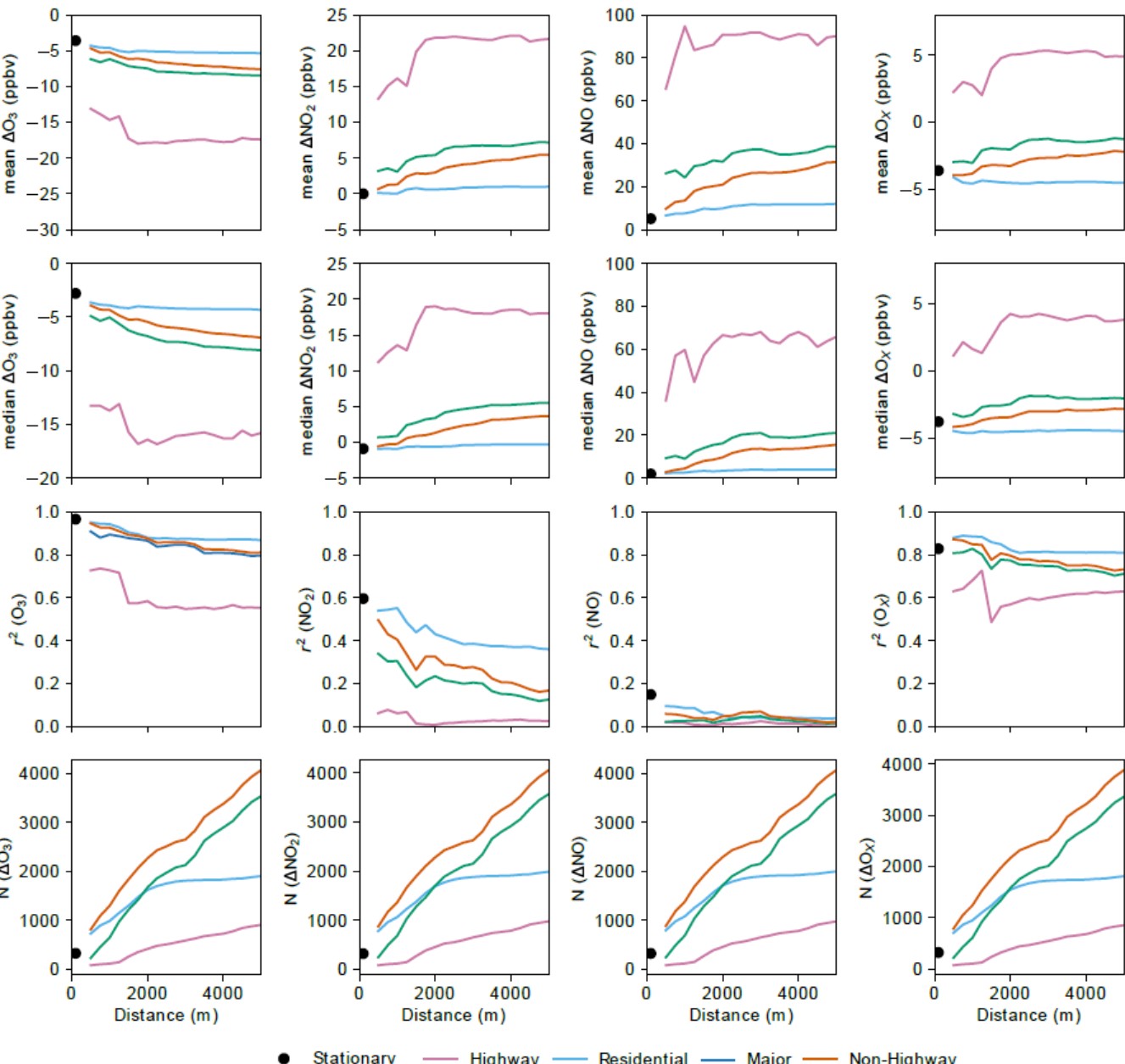

**Figure 3: Mean and median ΔX, coefficient of determination ($r^2$), and number of datapoints (N) for 1-minute car versus La Casa comparisons as a function of the maximum distance between the car and the La Casa site. Results from stationary collocations (Section 3) are shown as black dots, whereas different road classes are shown in different colors.**


For all road types, the bias between the mobile measurements and the reference site is lowest (and $r^2$ is highest) for distance buffers closest to the site. For the Residential road class, the bias between mobile collocation and parked collocation changes very little as the distance buffer increases for all species. The bias for observations collected on Major and Highway road types generally increases as additional samples are included at greater distances from the site.


When considering the results of bias by road type and distance class, it is important to note that the distribution of road type varies by distance. This is indicated by the number of data points (N) by road type as a function of distance in Figure 3. The La Casa site is in a residential area, so most of the roads within 500 m of the site are Residential. The designated drive patterns near the La Casa site resulted in most of the mapped roads within 2 km of the site being a combination of Residential and Major road types. Further away from the La Casa site, the roads consisted of a larger fraction of Major roads and Highways that were used to commute between the different areas where the denser mapping occurred.


For both Major and Residential road types, the bias and $r^2$ values tend towards the values from the parked collocations as the buffer distance decreases in most cases. Note that most of the mobile-to-stationary data within 1 km of the La Casa site was measured on the same days and generally within a 2 hour window of the stationary collocation data. For NO and $NO_2$, there are some slight differences between the parked collocation results and the 500 m buffer distance, with slightly higher bias on Major roads compared to the parked collocations. For NO and $NO_2$, this indicates an increase in the systematic spatial bias when in motion on nearby Major roads compared to when parked along with minimal differences in random variability. These results highlight the potential of in-motion mobile collocation compared to parked collocations in reducing the random variability, especially on Residential roads in the immediate vicinity of the stationary site. In general, however, there is remarkable consistency between the in-motion and parked collocation results for all pollutants when Highways are removed from the data set. Depending upon the target quality assurance guidelines of the study, there might be significant advantages to using in-motion mobile collocations instead of parked collocations to determine changes in mobile versus stationary measurement biases.




Scatterplots of the one minute mobile platform measurements versus the one minute La Casa measurements are shown in the Supplemental Information for $O_3$ (Figure S6), $NO_2$ (Figure S7), NO (Figure S8), and $O_X$ (Figure S9). These are shown, along with ordinary least squares linear regression statistics, for each road class individual and for distance buffers of 100 m, 300 m, 1000 m, 3000 m, and 10000 m. As with the stationary collocation scatterplots (Figure 2), these show the best agreement (i.e., closest to the 1:1 line) for $O_3$ and $O_X$, with moderate agreement for $NO_2$ and the worst agreement for NO. As suggested


above, the regression statistics (slope and intercept) appear to be a poor indicator of agreement, especially for directly emitted

species like NO (and, to a lesser degree, $NO_2$). Therefore, we focus our analysis on the central tendency metrics of the bias and the $r^2$ values as shown in Figure 3.

One final consideration for interpreting the impact of spatial variability in the Denver dataset is the impact of temporal aggregation. The availability of data from a fixed reference site at 1-minute time resolution was unique to the experimental

study in Denver, with additional instrumentation added to support the research objectives of the 2014 DISCOVER-AQ experiment. Data from regulatory monitoring stations in the United States will typically only be available at 1-hour time resolutions, and this is the data we had available for the California data set in Section 5. For mobile collocations to be broadly applicable to large-scale mobile monitoring applications, it is important to assess how the results change when the mobile data is compared with 1-hour stationary data. We averaged the mobile data by taking the mean of 1-second measurements within

each hour period that fit the appropriate road type and buffer distance criteria. Figure 4 compared the results of the 1-hour aggregated comparisons to the 1-minute comparisons as a function of buffer distance for the Non-Highway road class. Generally, the results for the 1-minute and 1-hour aggregation are similar in terms of both bias and $r^2$. For $O_3$ and $O_X$, the differences are minor. For $NO_2$ and NO there is a slight increase in the median bias as well as an increase in $r^2$ for the 1-hour comparisons versus the 1-minute comparisons. The $r^2$ at 1 hour for $NO_2$ and NO is variable as a function of distance buffer,

likely due to the limited size of the data set and the variable distribution of road type with distance, so this may not be an accurate assessment. The increase in $r^2$ for $NO_2$ and NO suggest that there is a modest reduction in the impact of random spatial variability at hourly aggregations; however, the increase in bias suggests that there is additional apparent bias as a result of aggregating. This could be due to the incomplete hourly aggregates from the mobile platform being compared to the full hourly observations from the stationary site. However, the Denver dataset is too limited to explore this hypothesis adequately.

The number of data points per hourly mobile collocation will be explored more thoroughly in Section 6.

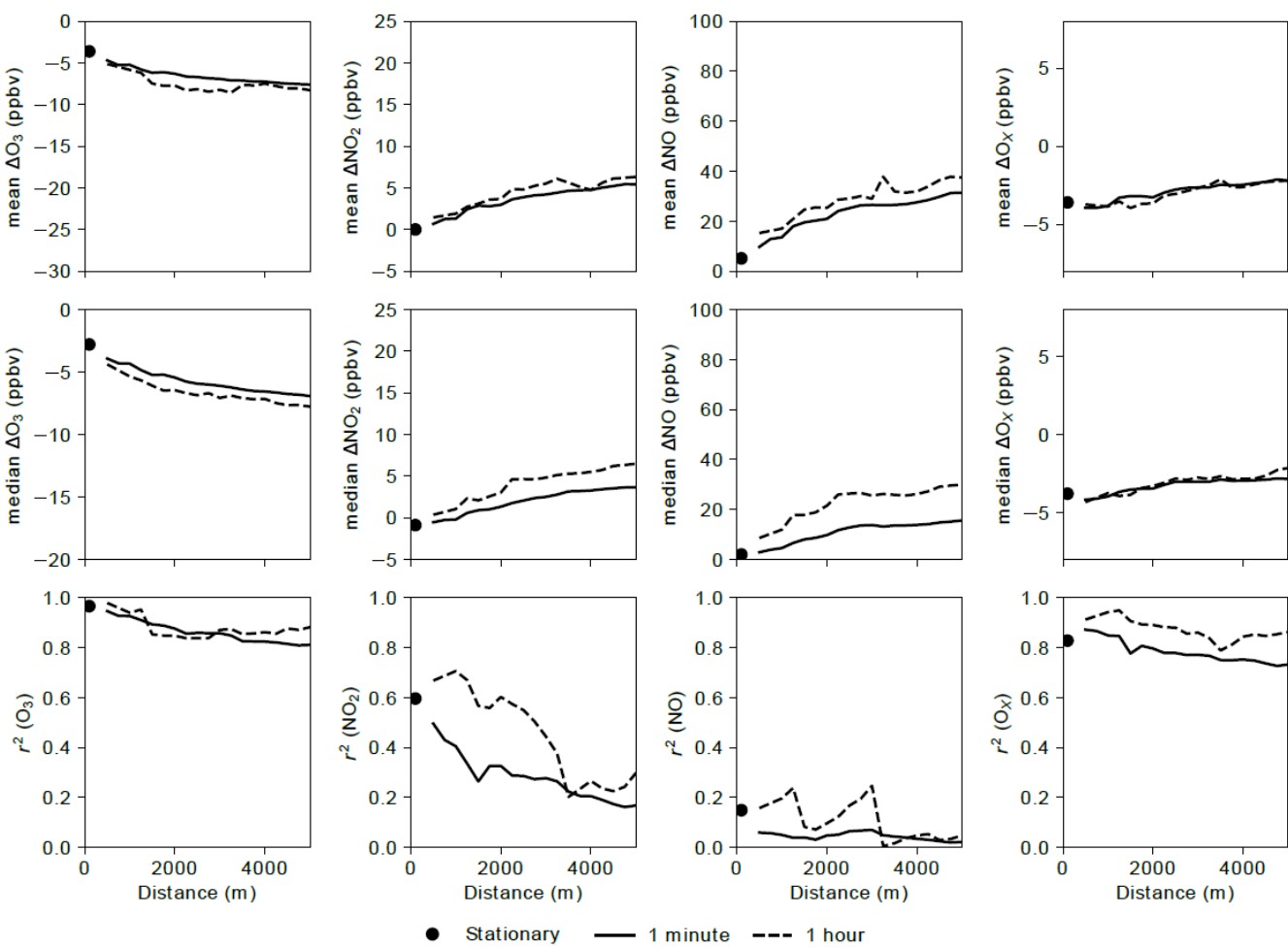

**Figure 4: Mean and median ΔX and coefficient of determination ($r^2$) for 1-minute and 1-hour car versus La Casa comparisons as a function of the maximum distance between the car and the La Casa site. Results from stationary collocations (Section 3) are shown as black dots. All Non-Highway roads are included in this analysis.**


## 5 Mobile platforms driving around fixed reference sites in California (2019 – 2020)

### 5.1 Methods

Aclima-operated fleet vehicles are equipped with a mobile sensing device, the Aclima Mobile Node (AMN), which measures carbon monoxide, carbon dioxide, $O_3$, NO, $NO_2$, $PM_{2.5}$, and total VOC. The AMNs are calibrated using the Aclima Mobile Calibration Laboratory (AMCL), a gasoline-powered Ford Transit van equipped with laboratory-grade air pollution measurement instrumentation. The AMCL was driven around the San Francisco Bay Area of California to calibrate the sensors

within the AMNs through comparison of the AMN sensor response with the laboratory-grade equipment collocated in the same van. The laboratory-grade instrumentation in the AMCL are calibrated regularly using reference gases to maintain bias and precision objectives. The calibration procedures have been described in Solomon et al. (2020). Bias and precision results

across approximately 25 zero and span checks are shown in Table 2. At the average concentrations observed during the study (11.9 ppbv for $NO_2$, 31.1 ppbv for $O_3$, and 11.9 ppbv for NO) the bias and precision in the span translates to less than 1 ppbv for all three pollutants.

As part of the validation process, the AMCL regularly drives around several Bay Area Air Quality Management District (BAAQMD) regulatory monitoring sites (Figure 5). These BAAQMD sites are equipped with EPA-approved FRM

and FRM measurements of NO, $NO_2$, and $O_3$ (or a subset of these species), in addition to other pollutants. We present an analysis comparing the AMCL reference measurements with BAAQMD reference measurements between November 2019 and October 2020. All measurements within 10 km of the regulatory sites were included in the data set used for analysis, as displayed in Figure 5.

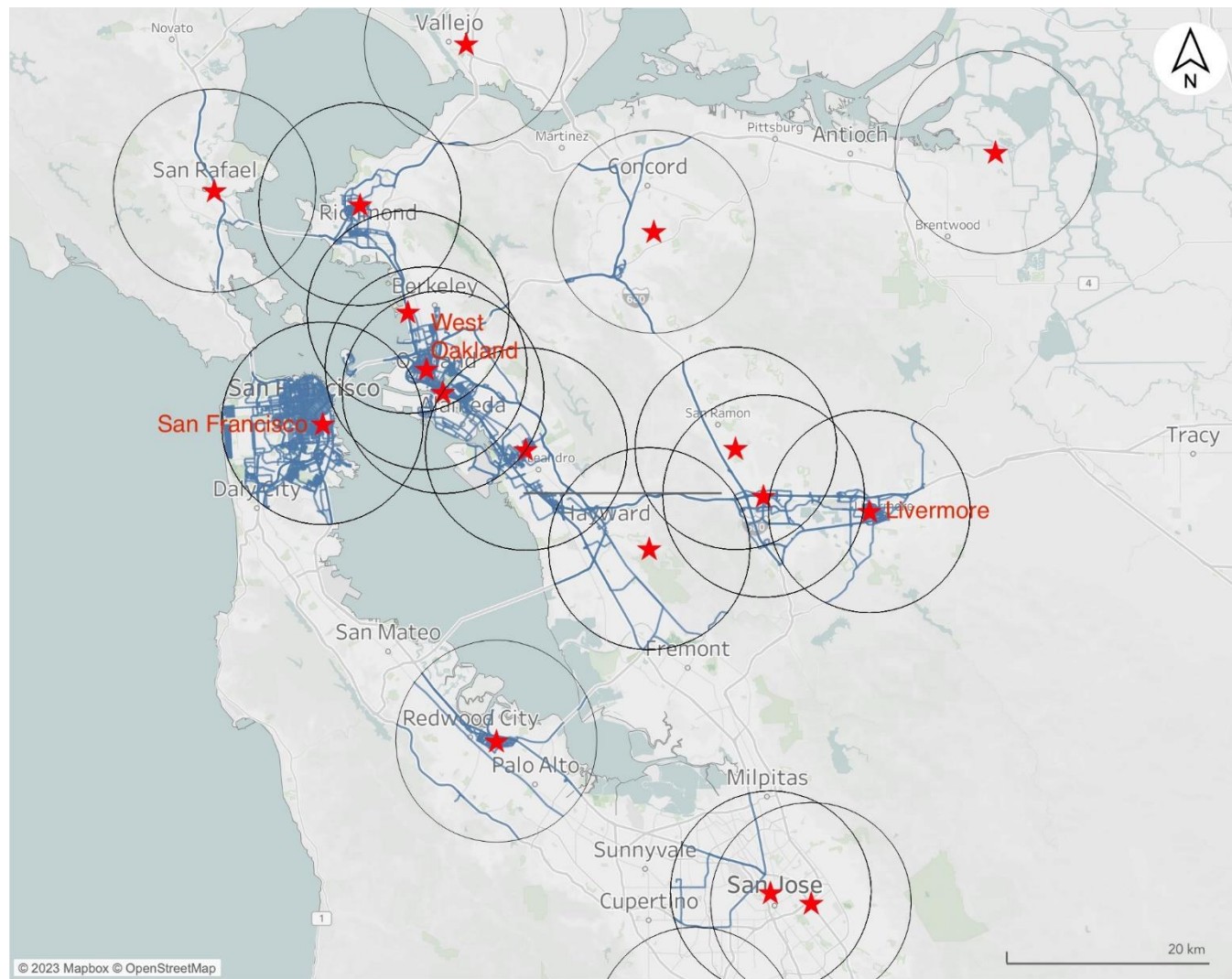

**Figure 5: Driving patterns around regulatory sites (red stars) in the San Francisco Bay Area. The circles delineate a 10 km radius around each regulatory station. The roads within each circle shown in blue are roads with measurements used in the analysis.**


**Table 2:** Precision and bias of measurements in the AMCL from approximately 25 quality assurance (QA) checks during the San Francisco Bay Area study period. Each QA check consisted of a zero and span point for each pollutant.

| Pollutant | Bias | | Precision | |
|---|---|---|---|---|
| | Zero | Span | Zero | Span |

| | | | | |
|---|---|---|---|---|
| NO$_2$ (ppbv) | 0.3 | 5.6% | 0.2 | 4.3% |
| NO (ppbv) | 0.7 | 4.4% | 0.3 | 4.4% |
| O$_3$ (ppbv) | 0.5 | 2.5% | 0.4 | 2.0% |

The dataset included comparisons to 19 different BAAQMD sites that represent several spatial representativeness

scales (40 CFR 58 Appendix D). Most of the data was collected near the Livermore, San Francisco, and West Oakland sites. These three regulatory stations are specifically labelled in Figure 5, and details of the road locations and road types mapped around each site are shown in Figure 6. Aclima selected these three stations to represent different climatological and land use regimes in the San Francisco Bay Area. The San Francisco station is in a warm summer Mediterranean climate with marine influence, cool winds and fog in summer, little overall seasonal temperature variation, and mixed residential with light

industrial land use. The Oakland station has a warm summer Mediterranean climate with marine influence, overnight fog in the summer, and mixed industrial and residential land use. The Livermore station has hot summer Mediterranean climate, inland with some marine influence, and predominantly residential land use while being upwind of a large fraction of the urban Bay Area emissions. In contrast to the mapping performed in Denver, measurements near these three monitoring sites generally involved mapping a significant fraction of the roads near the site (Figure 6). Measurements near other regulatory sites are also

included and were generally chance encounters due to the AMCL driving past these sites on its way to its daily mapping assignments, as illustrated in Figure 5. As a result, this data tends to be from Highways or Major roads and not mapped as comprehensively on a street-by-street basis.

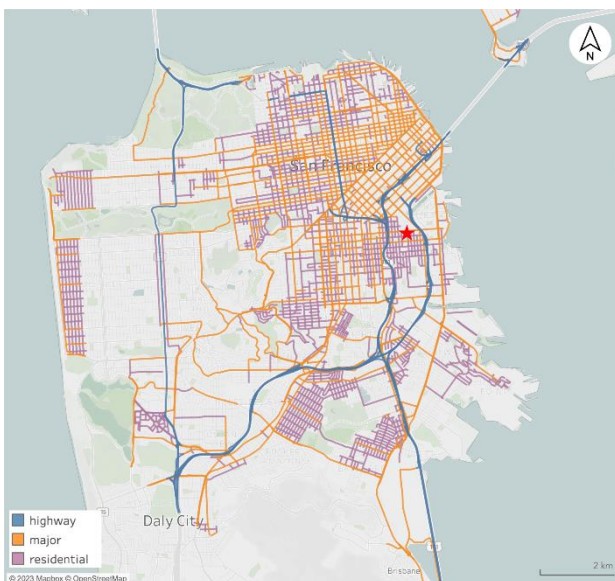

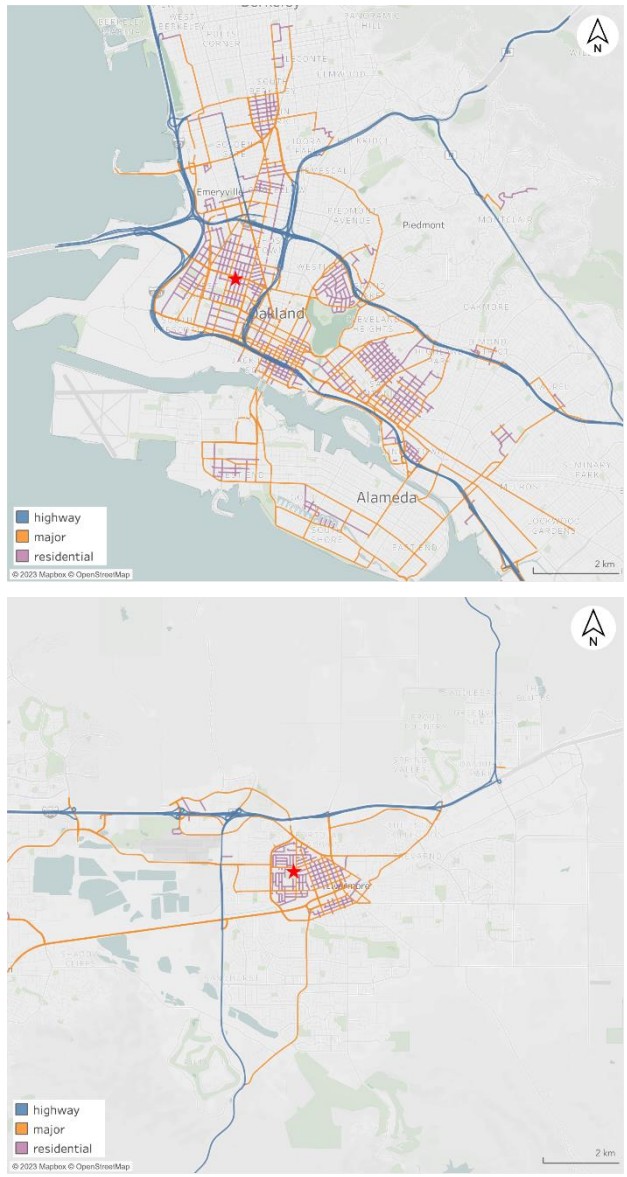

**Figure 6: Detail of the road locations and road types mapped by the Aclima Mobile Calibration Laboratory around the San Francisco (top), West Oakland (middle), and Livermore (bottom) regulatory sites (indicated by red stars) in the San Francisco Bay Area. Residential roads are shown in purple, major roads are yellow, and highways are blue**

Based on results from the 2014 Denver dataset (Section 4), we focused our analysis on two road type scenarios – Residential roads and Non-Highway roads. Because the BAAQMD measurements, like most regulatory gas-phase measurements in the USA are reported at 1-hour time resolution, we aggregated each subset of AMCL measurements up to one hour using the median as an aggregating function. We chose to use the median (versus the mean) to minimize the influence of outliers caused by local traffic emissions. This is in contrast to Sections 3 and 4 where the mean was used to aggregate the

one second data up to one minute or one hour. In general, using the median versus the mean produce similar results for $O_3$,

$NO_2$, and $O_X$; however, using the hourly medians versus means significantly reduces the impact of high NO outliers (peaks) on the NO aggregation. The fraction of each 1-hour collocation period that included measurements fitting the defined criteria varied depending upon the buffer distance and road type subset (Figures S10 and S11). For smaller distance buffers (e.g., 100 m or 300 m) or more restrictive road subsets (e.g., the Residential subset), the distribution was skewed towards a smaller fraction of measurements within each hour fitting the criteria for the comparison. For distance buffers 1 km and higher on

Non-Highway roads, the distribution was approximately uniformly distributed in the 0 – 100% range. No minimum number of data points were required in each hourly average, such that any individual hourly aggregate may include anywhere between a few seconds and a full hour of 1-Hz mobile platform data.

## 5.2 Results and Discussion

        Using the approach described in Section 4.2, both the mean and median bias values and $r^2$ values are shown as a

function of road type and distance buffer from the fixed regulatory sites in Figure 7. Similar to Section 4.2, we associated the central tendency of the bias to reflect both instrument biases as well as persistent spatial biases and the $r^2$ values to reflect random spatio-temporal variability. The California study is bolstered by a much larger data set (note that N in Figure 7 represents the number of hourly aggregates, whereas the N in Figure 3 represents the number of 1-minute aggregates). This dataset was also collected over a full year and includes comparisons with multiple stationary sites. Despite the different

geographic locations and scale of data collection between the California and Denver studies, the general patterns observed in Figure 7 are highly consistent with the patterns observed in Figure 3. For example, higher magnitude of biases and lower $r^2$ at larger buffer distances and significantly worse agreement for Highways than other road classes. The California results do show somewhat higher $r^2$ and smaller magnitude of biases in general than the equivalent hour-averaged results for the Denver study (i.e., the hourly traces in Figure 4). The higher $r^2$, in particular for $NO_2$, could be due to the larger dataset used, collected over

a full year of atmospheric and climatological conditions at multiple sites. This led to both a more representative dataset and a wider range of sampled concentrations than the one-month, single-site Denver analysis done in Section 4. The smaller bias, for $O_3$ and $O_X$, could be due to better inter-lab comparability in the California dataset, but aggregating data across multiple sites may also explain a reduction in the systematic bias. For example, if one site has a slightly positive bias and another site has a slightly negative bias (due to monitor siting or site-to-site calibration variability), those biases will partially cancel each

other out. Variances in traffic patterns, road type distributions, and other factors could also influence differences in biases in different geographic regions or using different driving patterns, so the range of biases must be measured for each individual study region and study design.

        Figure 7 also shows close agreement between the results for the Major roads and the Non-Highway roads, reflecting the large number of Major roads (versus Residential roads) included in this study compared to in Denver. It is interesting to

note that the number of datapoints (N) is almost identical for the Major and Non-Highway road classes. This is because most hour periods that included driving on Residential roads also included driving on Major roads, so those hour periods were

counted separately for the separate Residential and Major road classes but only one time for the Non-Highway road classes. Unlike the Denver dataset, this dataset shows remarkable consistency in the $r^2$ values between the Residential, Non-Highway, and Major road class subsets, suggesting that large scale application of these comparisons provide similar random spatial biases ($r^2$ values) for Residential and Non-Highway roads. As with the Denver results, there is also minimal variability in these metrics with increasing distance buffers up through 3000 m (and higher).

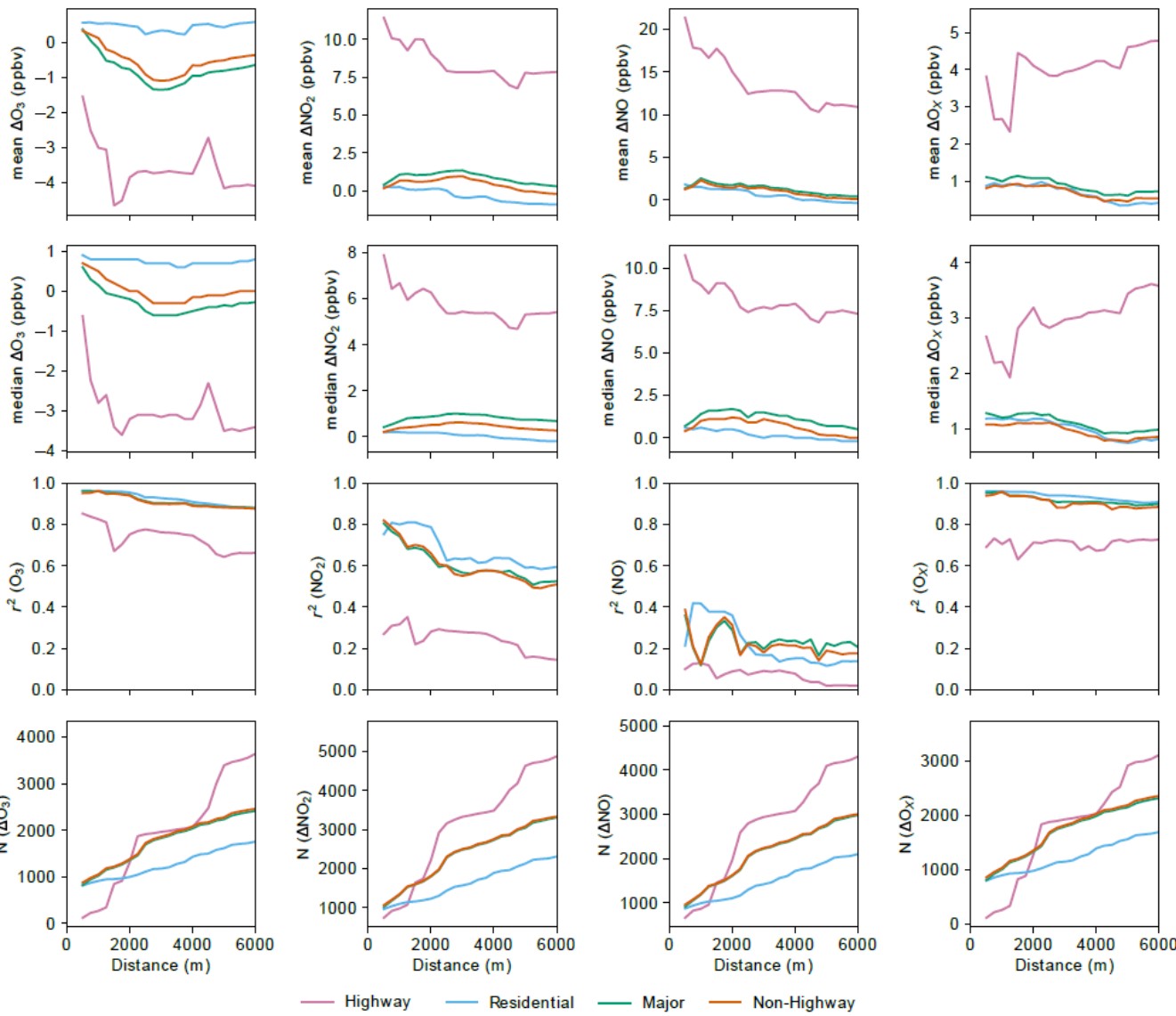

**Figure 7: Mean and median ΔX, coefficient of determination ($r^2$), and number of datapoints (N) for 1-hour AMCL versus regulatory site comparisons as a function of the maximum distance between the AMCL and the regulatory site.**

 **Depending upon the buffer distance, the comparisons can include data for up to 19 BAAQMD regulatory sites. Different road classes are shown in different colors.**

## 6 Operationalization of mobile-versus-fixed-site comparisons for ongoing instrument assessment

These results in both Denver and California provide a blueprint for how operational decisions can be made for efficiently collecting collocation data to determine systematic measurement bias while accounting for the tradeoffs between the rate of data collection and uncertainty tolerance for attributing changes in $\Delta X$ to measurement bias. The optimal approach will balance these tradeoffs in a way that maximizes N, maximizes $r^2$ (i.e., minimizing random spatial variability), and minimizes the spatial bias component of $\Delta X$. Based on our results in both Denver and California, it is advantageous to remove Highway road segments from the data set, resulting in a decrease to both the random and the persistent spatio-temporal variability with minimal cost on the number of data points collected. Although more complex peak-removal algorithms can achieve similar goals, they add complexity without necessarily improving the comparison and add may additional arbitrary bias (e.g., "cherry-picking") to the resulting comparisons. A major driving factor for optimization of the buffer distance and which non-Highway road types to include is likely to be the rate with which sufficient number of collocation data points can be collected. For example, consider two end points for this problem: 1) only Residential roads with a buffer distance of 500 m; and 2) all non-Highway roads with a buffer distance of 3000 m. While the non-Highway and large buffer distance would result in a slightly increased bias and random variability over the Residential only and small buffer distance scenario, it also is a larger dataset by a factor of 2-3 and would allow for simultaneously mapping a larger area to meet the monitoring objectives more efficiently. Therefore, understanding the impact of the size of the data set is a critical first step for understanding how to design an operational strategy for using collocations to assess measurement bias.

Quantifying the impact of the size of the data set requires an analysis along an additional dimension that has not yet been considered - the uncertainty with which $\Delta X$ can be determined. While there is likely a close relationship between the magnitude and the uncertainty of $\Delta X$, the uncertainty in $\Delta X$ is more important than the absolute value of $\Delta X$ in the context of using mobile to stationary comparisons to attribute changes in measurement bias over time. Assuming that random spatial variability is the primary source of error in determining the true systematic spatial bias, the rate at which collocation data can be collected becomes a factor that may need to be weighted more heavily than the absolute magnitude of the spatial bias. This will be especially true for cases where $r^2$ is expected to be low (i.e., for $NO_2$ or $NO$), which indicates a higher degree of random spatial variability and, therefore, requires additional data collection to achieve an equivalent uncertainty reduction in $\Delta X$. In the following section, we quantify the uncertainty in $\Delta X$ as a function of the amount of data collected and aggregated.

### 6.1 Using running median bias values to identify instrument issues

Here we explore how the amount of data aggregated over a specified amount of collection time impacts the instrument bias calculated for the instruments that we know have a stable calibration over time. Our objective is to find the minimum

temporal window required to determine reasonable uncertainty bounds for the determination of measurement bias. For the analysis, we use the California dataset of hourly $\Delta X$ values and calculate a running median of hourly median $\Delta X$ as a function of the number of hours of data in the running average, N. We use median $\Delta X$ as the results are less affected by outliers resulting from local emission plumes and thus produce estimates of the bias with smaller magnitude. For this analysis, we focus on a buffer distance of 3000 m and all Non-Highway roads, which results in a sufficiently large dataset to perform this analysis and also represents what may be a more commonly encountered scenario in complex urban environments with varied emission sources than would be simulated by a narrower buffer distance and solely Residential road types form this data set. Figure 8 shows the minimum and maximum bounds for observations of the running median $\Delta X$ for the entire dataset as a function of window size N. Note that each individual one-hour $\Delta X$ is the median of one-second differences during that one hour time window. Therefore, we are taking a running median (over individual hours) of hourly median $\Delta X$ values. The range between the upper and lower traces in Figure 8 provides an estimate of the uncertainty in median $\Delta X$ due to random spatial variability, and thus a measure of the magnitude of change in systematic measurement bias that we can expect to be observable by this approach. For each running median window size N, the upper and lower traces reflect the maximum and minimum of the set of running N-hour medians from this dataset. As expected, the range of values decreases with increasing window size as the influence of random spatial and temporal variability on the calculated bias is reduced. At around 30 to 40 hour window size, the range between the minimum and maximum stabilizes and does not reduce appreciably with further aggregation.

The number of 1-second data points contributing to each 1-hour $\Delta X$ value in this analysis is highly variable (Figures S10 and S11). For data within a 3 km buffer distance, the number of 1-second data points in each hour aggregate ranges from just a few seconds to a full hour and is fairly evenly distributed. The time-resolved data from mobile mapping, by its nature, can have significant temporal and spatial variability and using only a few seconds of data to compare to regulatory data with hourly time resolution may result in additional noise and perhaps invalid assessments of instrument bias in the analysis. To explore the degree to which including hourly aggregates with only a few seconds of data impact the resulting analysis, we repeated the calculation with a restriction on the amount of mobile data necessary during each hour period for inclusion in the analysis. We used completeness criteria of 5 and 10 minutes, which would require at least 5 minutes (or 10 minutes) of valid data during each hour period for that period to be included in the sample data set for analysis. Note that these 5 or 10 minutes need not have been consecutive. Results are shown alongside the base case (no time-base restriction) in Figure 8. While there are only subtle differences in the results between the three completeness criteria, the use of 5 or 10 minute minimum cutoff for each hourly aggregation does result in the convergence of median $\Delta X$ values at somewhat smaller window sizes, particularly for $O_3$. However, atby a rolling window size of around N = 40 hours or higher, the improvement is marginal (e.g., less than a fraction of a ppbv) in most cases. This analysis has an important implication for the design of a mobile data collection plan that incorporates monitoring in the vicinity of stationary monitors for quality assurance purposes while simultaneously meeting mobile monitoring objectives. The results in Figure 8 imply that spending only 5-10 minutes, or even less, within 3 km of a monitoring site can be an effective collocation data point, and the more critical parameter is the number of distinct hourly comparisons. This minimizes the effort needed to collect 40 distinct hours near the monitoring site and, thus, the impact on

mobile data collection efficiency in areas of interest farther away from stationary monitors.For the California data used in this analysis, the range of $\Delta X$ values of around $\pm 4$ ppbv for $O_3$, $NO_2$, and $O_X$ and $\pm 8$ ppbv for $NO$ represent the minimum instrument drift we can expect to detect using this method. This is determined as the range between the upper and lower traces in Figure 8 for each pollutant. We find that it is possible to detect this magnitude of instrumental drift over the time with which it takes
to collect approximately 40 distinct hourly collocation data points. The time it takes in practices to acquire a 40 hour median $\Delta X$ depends heavily on the specific data collection plan. For our dataset, we have ~1600 (for $O_3$) to ~1900 (for $NO_2$) hourly datapoints in our 3 km, Non-Highway dataset from one year of driving in California. For a rolling window size of 40, this gives an "effective response time" on the order of one week (7 to 9 days), where the response time is the study period (365 days) divided by the number of discrete hour periods (1600 to 1900)  times the rolling window size (40). This response time
is likely atypical, since the objectives of this monitoring plan was specifically to collect data in close proximity to a predetermined set of stationary monitors. On the other hand, most typical mobile monitoring deployments would aim to collect over a much broader area and there would be much higher value in collecting in areas farther removed from where stationary monitors exist. Nevertheless, it does provide a framework for planning purposes. For example, if it is desirable to detect instrument drift over a quarterly time period, the mobile collection plan would need to incorporate about 3 separate visits per
week to Non-Highway roads within a 3 km radius of a stationary monitor. Importantly, these  "visits" do not need to be exclusively 1-hour long visits, and visits of only 5 – 10 minutes or even a few seconds can be viable. While imposing a minimum data completeness criteria might improve the range of running median $\Delta X$ values at lower N values, it comes at the cost of a reduction in the amount of data and thus an increase in the "effective response time" (for similar values of N).


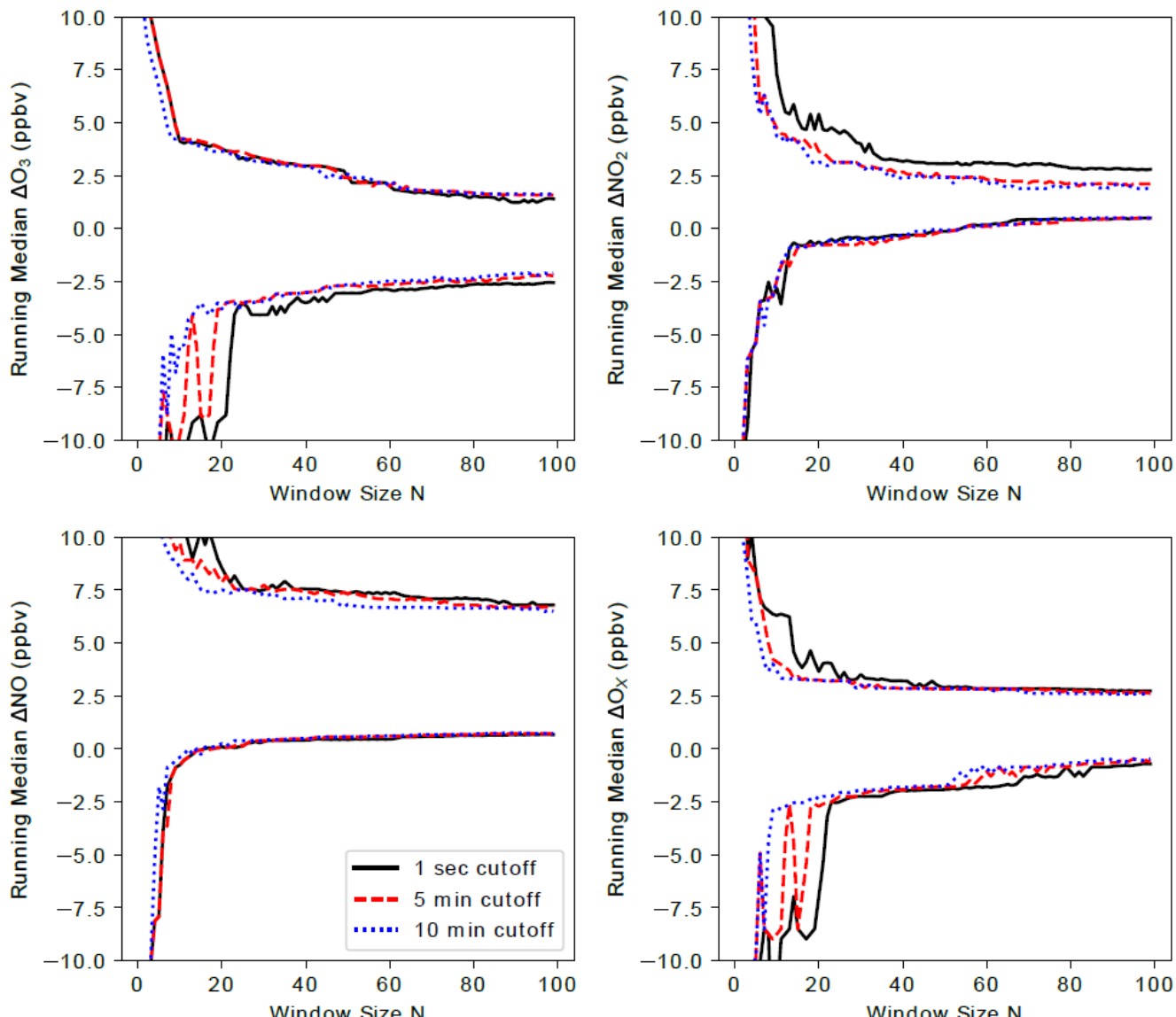

**Figure 8: Influence of window size on the range (maximum and minimum) of running median $\Delta O_3$, $\Delta NO_2$, $\Delta NO$, and $\Delta O_X$ values for all Non-Highway roads within 3 km of a fixed reference site. We show the relationship for three "minimum data" cutoff values – 1 second, 5 minutes, and 10 minutes – which define the minimum length of valid data necessary during the hour-long averaging period necessary to include that hour in the sample dataset.**

Using a running 40 hour time window, we show a timeseries of hourly median $\Delta O_3$, $\Delta NO_2$, $\Delta NO$, and $\Delta O_X$ in Figure 9. Both the raw 1-hour data (red points) as well as the 40-hour running median (black line) are included. While there is a large degree of variability for any individual $\Delta X$ value, the running median reduces that random variability and provides an estimate with quantified uncertainty bounds that can be used to identify drift. For long-term driving campaigns that frequently pass near

stationary monitoring sites, this method appears to be a practical way to monitor systematic bias in mobile measurements in an ongoing basis. In Section 6.2, we apply this method to an example of two $NO_2$ sensors in Aclima's mobile fleet to show an example of how this method could identify real drift in lower-cost sensors.

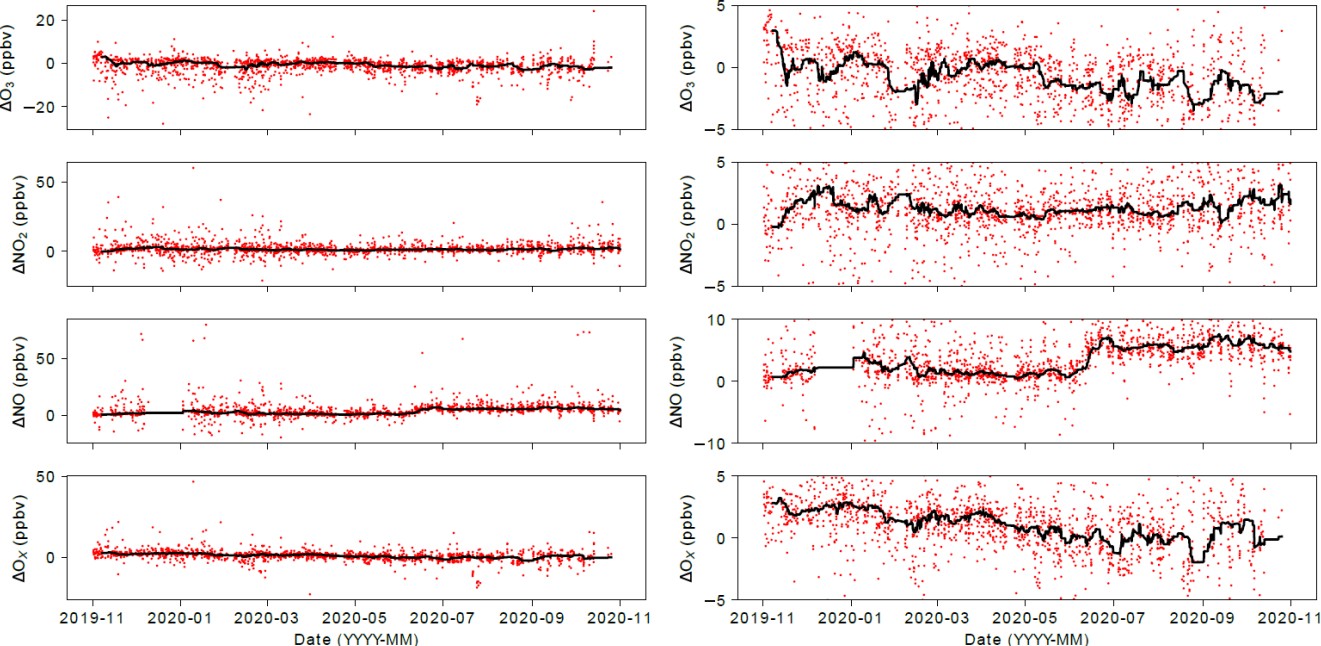

**Figure 9: Timeseries of 3 km $\Delta O_3$, $\Delta NO_2$, $\Delta NO$, and $\Delta O_X$, showing individual 1-hour measurements (red points) and 40-hour running medians (solid line). The right column truncates many of the 1-hour measurements and scales the y axis to highlight the range of the running median values.**


## 6.2 Case study: $NO_2$ sensors in Aclima's mobile fleet

To illustrate how the running median method can be used to identify drift in sensors deployed for mobile monitoring, we show an example using $NO_2$ sensors deployed as part of Aclima's mobile collection fleet. We analyzed mobile versus stationary differences for two $NO_2$ sensors deployed in two different vehicles across a multi-month deployment in California. In contrast to the AMCL results shown in Section 5 and Section 6.1, these sensors are not regularly calibrated during their mobile deployments. Instead, they are calibrated initially via collocation with the reference instruments in the AMCL (prior to deployment) and calibrated a second time in the AMCL at the end of their deployment period. For Sensor A, the original calibration was found to have held well during the post-deployment calibration check, with a mean bias of +2.2 ppbv. This was well within our ±6 ppbv acceptance criteria for the $NO_2$ sensor. Sensor B, on the other hand, was found to have a mean


bias of -15.5 ppbv during its post-deployment calibration check, indicating significant drift beyond what we consider acceptable performance.

     While the sensors are deployed, the only possible in-situ evaluation of the calibration of these sensors was using mobile-versus-stationary comparisons with fixed reference sites. Over the course of the deployment for both vehicles, they made frequent passes within 3000 m of various regulatory sites. These sites were not specifically targeted – rather these

encounters occurred over the course of typical data collection for hyperlocal air pollution mapping. The hour-averaged results for each of these sensors is shown in Figure 10, along with the 40-hour running median. The in situ running median biases compared to the regulatory sites is consistent with the post-deployment bias determinations, indicating that mobile versus stationary comparisons could be used to detect drift in mobile NO$_2$ sensors while deployed. In addition, it is apparent in Figure 10 that Sensor B had significant discrepancies compared to the fixed reference sites that were outside of the expected ±6 ppbv

uncertainties. Therefore, this method could have been used to identify potential issues with Sensor B during deployment without waiting until a post-deployment calibration. While a more detailed analysis across multiple devices and deployments would be required to establish this approach as an accepted method, this case study demonstrates how such an approach might be feasible.

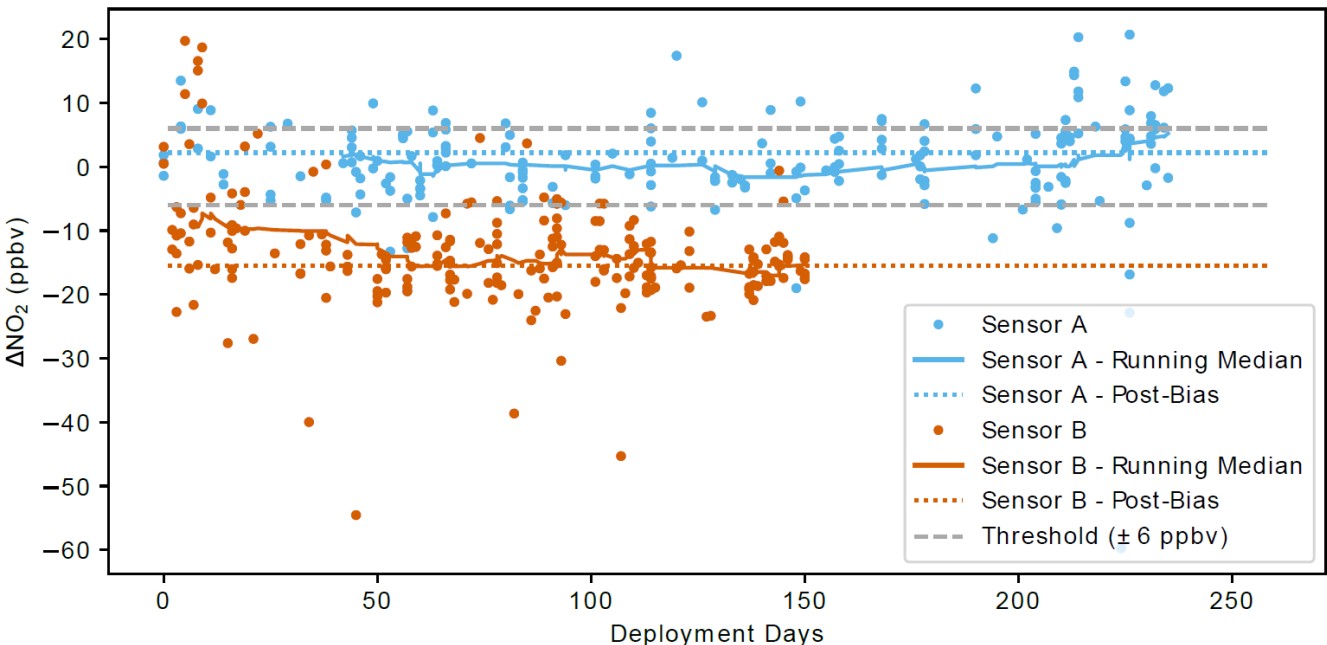


**Figure 10: Timeseries of 1-hour $\Delta NO_2$ and 40-hour running medians for two sensors (Sensor A and Sensor B) deployed as part of Aclima $NO_2$ sensor deployment. Uncertainty threshold (±6 ppbv) and post-campaign sensor biases also shown.**

## 7 Spatial heterogeneity and implications for applicability to additional pollutants

In this manuscript, we focus on three major pollutants ($O_3$, $NO_2$, and NO) with very different behaviors in the atmosphere. $O_3$ is predominantly a regionally-distributed secondary pollutant with high background concentrations and negative deviations in direct emission plumes, especially those containing NO (which rapidly titrates $O_3$). $NO_2$ is both a primary and secondary pollutant that has moderate regional background concentrations and falls into the category of a co-emitted pollutant with high regional background (Brantley et al., 2014), along with species like $PM_{2.5}$ and $PM_{10}$. NO is a

primary pollutant with a short lifetime, especially in the presence of $O_3$, and has high peak concentrations and low background concentrations. NO falls into the category of a co-emitted pollutant with low regional background (Brantley et al., 2014), which also includes pollutants such as carbon monoxide, black carbon, and ultra-fine particles.

        In addition to providing valuable insights into deployed sensor data quality, our analysis also provides information about the spatial heterogeneity of these atmospheric pollutants and the spatial representativeness of measurements at stationary

sites. If we consider the coefficient of determination ($r^2$) of the mobile vs stationary regressions to be a simple proxy for spatial homogeneity (setting aside for now the important temporal component to this variability), with higher $r^2$ indicating a more spatially homogeneous pollutant, then we conclude that $O_3$ is more spatially homogeneous, NO is more heterogeneous and $NO_2$ is between the two. This is consistent with our understanding of emission sources and atmospheric lifetimes of these different species under typical urban conditions. The temporal component of this spatial variability is, of course, a key

consideration and, as such, the linear regressions described by the $r^2$ values (and shown in Figure S12) are sensitive to the spatial variability within an hourly snapshot, while the relationships shown in Figure 8 describe the spatial variability over different temporal aggregations (i.e. the median window size). The field of hyperlocal air quality monitoring is predicated on the fundamental principle that aggregating many samples over time is required to reduce the impact of temporal variability to observe persistent spatial trends in concentration (Apte et al, 2017; Messier et al., 2018; Van Poppel et al., 2013). While the

hourly variability between mobile and stationary measurements may be a reasonable proxy for spatial heterogeneity, aggregations over longer durations provide a more accurate measure.

        The high degree of correlation between hourly mobile and stationary measurements might suggest that $O_3$ has minimal spatial variability. However, our analysis shows that there are measurable spatial gradients in $O_3$. This is particularly true for Highways, and to a lesser degree Major roads, as compared to Residential roads and stationary sites. Figure 3 for example,

clearly shows observable differences in the mobile measurements in Denver compared to the stationary measurements both as a function of distance from the site and as a function of road type. For this reason, our analysis intentionally removes highways specifically to reduce the influence of local emission plumes in the data set so that the mobile to stationary comparison can be more readily indicative of measurement bias. However, the full data set could offer a means to more closely investigate the

interplay between the hyperlocal spatial patterns of these 3 closely related pollutants ($O_3$, $NO_2$, and NO) and the implications for effective emissions controls at hyperlocal scales.


This relationship between spatial heterogeneity, mobile vs stationary correlation coefficients, and the practicality of using mobile vs stationary comparisons as a quality control method suggests that it is possible to make some educated assumptions about how other pollutants would behave. For example, $PM_{2.5}$ has both primary and secondary sources and would likely show mobile to stationary correlations similar to $NO_2$, or possibly $O_3$ in some areas. Conversely, black carbon, carbon

monoxide, or ultrafine particles would likely behave more similarly to NO, given their variability in near-source areas results primarily from direct emissions. While we expect significant differences in correlation with stationary sites for different pollutants across the spectrum of spatial variability, the results in Figure 8 for NO, $NO_2$, and $O_3$ suggest that all pollutants would likely follow a similar pattern of decreasing range of $\Delta X$ with increasing median window size. The key differences that would be expected for different pollutants would be the optimal median window size (although the optimal median window

size is similar for all 3 measured pollutants in this study) and the width of the confidence intervals around the central tendency of the differences between mobile and stationary measurements, and thus, the magnitude of systematic measurement bias that could be detected.

## 8 Conclusions

In this paper, we address the issue of ongoing quality assurance during a large-scale mobile monitoring campaign, with a focus

on discerning changes in instrument performance over time during mobile platform deployment. To assess instrument drift over time using any sort of collocation or sensor-versus-reference comparison, it is first necessary to constrain the uncertainty inherent in the collocation or comparison process. We used a set of parked and moving mobile monitoring data from a one-month study in Denver, CO (2014) and compared reference grade NO, $NO_2$, and $O_3$ measurements from a mobile platform to fixed reference site measurements both when the mobile platform was parked side-by-side with the reference site and when

driving at distances out to several kilometers from the reference site. Using data from a more extensive, 1-year study in California (starting November 2019), we show large-scale comparisons of hourly mean mobile measurements to hourly fixed reference site measurements. We highlight the importance of grouping data based on street type to reduce the influence of emission plumes, which are most abundant on highways. We demonstrate a possible data aggregation technique for large-scale, long-term comparisons. Hourly averaged regulatory site data are reported by most state and local air quality monitoring

agencies in the US and many other countries. These hour-aggregated mobile platform measurements show excellent agreement with hour-averaged fixed site measurements using running medians of moving mobile versus fixed site differences. The work presented here will be extended in the future to examine how these methodologies can be used to assess the ongoing performance of low-cost sensor nodes in mobile monitoring platforms.

**Conflict of Interest**

The authors declare no conflicts of interest.

**Data Availability**

Data used for this manuscript is proprietary and owned by Aclima, Inc. (https://aclima.io). Interested researchers are encouraged to contact Aclima, Inc. for data availability and collaboration opportunities. Additional questions about analysis techniques or code can be directed to the corresponding author of this manuscript.

**Author Contribution**

Conceptualization – ML, SK, and PS
Data Curation – ML, BL
Formal Analysis – AW, ML, BL
Funding Acquisition – ML, BL, PS
Investigation – AW, ML, BL
Methodology – AW, ML, BL, and PS
Project Administration – ML, SK, PS
Resources –ML, BL, PS
Software – AW, ML, BL
Supervision – ML, SK, PS
Validation – ML, BL, and PS
Visualization – AW, ML
Writing – Original Draft – AW, ML, BL
Writing – Review and Editing – AW, ML, BL, PS

**Disclaimer**

This manuscript has been subjected to internal review at the U.S. Environmental Protection Agency and has been cleared for publication. Mention of trade names or commercial products does not constitute endorsement or recommendation for use. The views expressed in this article are those of the author(s) and do not necessarily represent the views or policies of the U.S. Environmental Protection Agency.


## Acknowledgements

The project was conducted in part under a Cooperative Research and Development Agreement between the US EPA and Aclima, Inc., EPA CRADA #734-13, April 13, 2013 and Amendment 734-A14 to April 13, 2023. The authors would like to thank the internal technical and management reviewers from the U.S. EPA, including Shaibal Mukerjee, Robert Vanderpool, Libby Nessley, Michael Hays, and Lara Phelps, for their excellent suggestions for modifications and improvements to the draft manuscript. The authors would like to thank US EPA Region 8 for support of the project in Denver and the cooperation of data sharing by DISCOVER-AQ and the Colorado Department of Public Health and Environment. We gratefully acknowledge the contributions of Davida Herzl and Karin Tuxen-Bettman for their vision and support, and Christian Kremer, Matthew Chow, Cassie Trickett, Marek Kwasnica, and the Aclima Technical Operations Team for the dedicated support of Aclima's data quality operations.

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
