# Peer review of "Mobile air quality monitoring and comparison to fixed monitoring sites for instrument performance assessment"

_Atmospheric Measurement Techniques, 2023_

## Referee Comment (RC2)

Reviewer comments for:

**Mobile Air Quality Monitoring and Comparison to Fixed Monitoring Sites for Quality Assurance**

This review is for the above manuscript submitted for publication in Atmospheric Measurement Techniques. The manuscript partially develops and proposes implementation of a new quality assurance procedure to evaluate changes in instrument performance in mobile monitoring of air quality. To do that, the authors use high-temporal resolution (O ~ 1s) mobile-monitoring data collected using regulatory-graded instruments from two campaigns conducted in different regions for very different lengths of time for three pollutants, O3, NO2, and NO. The authors then compare stationary referencing of this data during collocations with the regulatory monitor to the referencing of "vehicle-in-motion" concentrations with regulatory monitoring data (based on distance and road type from the regulatory monitor) for one site, and find similar performance evaluations for a regionally distributed pollutant O3 (r2 > 0.9), moderate performance for NO2 (r2 > 0.5), and poor performance for a primary pollutant, NO (r2 < 0.2) in their new approach. For the second site, the authors do not conduct stationary referencing and only perform the latter "vehicle-in-motion" referencing to the regulatory monitor to estimate optimal temporal "running windows" to identify instrument issues. They calculate that for a 3 km spatial window, a temporal running window of 40 hours for data would allow detection of a systematic measurement drift or sudden instrument or sensor malfunction over the time scale of 7-9 days. While the authors motivate this work to identify and address systematic measurement drift or malfunction, the same is are not demonstrated in their results nor do they show the implementation of this method on any dataset. Additionally, in its current format, the manuscript uses linear models even when they do not seem applicable, especially for NO and NO2. Finally, the deficient analysis of high-concentration plumes further limits the usefulness of the proposed method. I recommend that the authors significantly revise and resubmit this manuscript for further consideration.

**1. Deficient motivation and lacking significant findings:** The authors begin by motivating this work as to "provide an important tool for ongoing quality assurance during mobile measurement campaigns" [Line 51]. They believe that "through ongoing comparisons of fixed reference site and mobile measurements, it may be possible to identify instrument drift over time or changes in instrument performance that could indicate a malfunction" [Lines 47-49]. The other motivation the authors suggest is that ongoing "mobile-versus-fixed-site comparisons are more scalable than frequent site-by-side parked collocations", "which is particularly important during sustained, multi-vehicle (and fleet-based) mobile monitoring campaigns" [Lines 50, 57, 58]. However, in their findings, the authors conclude and I agree that **this method "is not an absolute method for calibration or instrument verification, as a direct collocated comparison with reference monitors is"** [Line 458]**.** In the two campaigns the authors conducted, they performed daily checks with zero and span gases and in one campaign, conducted direct collocation comparisons. These **standard approaches provide lab-grade confidence in the measurements including with regards to instrument drift and malfunction**

**and no replacement for them has been identified in this work.** While authors do find that regional pollutants are correlated strongly in "mobile-versus-fixed-site comparisons", these pollutants are expected to exhibit regional homogeneity and this result is not a significant contribution of this work. Frequently, measurements dominated by secondary pollutants are referenced to nearest (but far-located) reference monitors, both in stationary and mobile monitoring. **What the authors in fact demonstrate is the weakness of the "mobile-versus-fixed-site comparisons" for pollutants with high spatial variability such as NO.** While 15-16 stationary collocations only 20 mins each conducted over 2-3 weeks in the Denver campaign to calibrate against regulatory monitors yield mean r2 of 0.4, this comparison performance drops precipitously to r2 <= 0.2 in hourly averaged "mobile-versus-fixed-site comparisons" (Table 3). This is not surprising, since "spatial coverage from mobile monitoring reveals patterns missed by the fixed-site network", especially for primary pollutants (Chambliss et al., 2020). Anyway, scaling their standard stationary collocations up to a year (approximately equivalent to the length of the California campaign, also presented here) totals to about 60 hours. In this work, the authors propose sampling in their newly developed approach within a crude spatial scale of 3 km for about 40 rolling hours to identify instrument drift/malfunction. Clearly, there is little advantage to switching to this new approach given the drop of ~50% in the measure of performance in the "mobile-versus-fixed-site comparison" in the only primary pollutant monitored compared to the standard "stationary collocation". In short, while the authors argue that stationary collocations "ensures comparability only at that specific location and only under the specific atmospheric conditions over which the collocation occurred" (Lines 37-38), **they have demonstrated that stationary collocations perform significantly better than their proposed method.**

**2. Use of linear models:** As Figures 1 and 4 demonstrate, a linear model seems insufficient to compare measurements of NO and NO2. There is a clear baseline effect, where only a small fraction of variance in concentrations can be explained by variations in the reference site. The extensive dependence on presentation using linear models in the manuscript for the Denver phase further weakens this manuscript. I suggest the authors' reconsider the presentation approach as well as the feasibility of a "mobile-versus-fixed-site comparison" given visible baselines that are developed primarily based on comparison with the Denver "stationary collocations".

3. **Studying outlier plume events**: Hyperlocal monitoring has a lot of value given its ability to map pollutant exposures, especially plumes of primary pollutants with high spatial variability. However, prior work suggests that primary pollutant spatial patterns can be predicted well using land-use regressions (Robinson et al., 2019). In contrast, regionally distributed pollutants are not well represented by such regressions but are relatively spatially homogeneous and can be estimated using regional monitoring (Shah et al., 2018). Given these well-established priors, the baseline effect in (2) above, the high-concentration plumes or "outliers" in primary emissions should be studied separately. Similar mobile monitoring work has been done previously and could be referenced (Robinson et al., 2018).

1. Chambliss, S. E., Preble, C. V., Caubel, J. J., Cados, T., Messier, K. P., Alvarez, R. A., LaFranchi, B., Lunden, M., Marshall, J. D., Szpiro, A. A., Kirchstetter, T. W., and Apte, J. S.: Comparison of Mobile and Fixed-Site Black Carbon Measurements for High-Resolution Urban Pollution Mapping, Environ. Sci. Technol., 54, 7848–7857, https://doi.org/10.1021/acs.est.0c01409, 2020.

2. Robinson, E. S., Gu, P., Ye, Q., Li, H. Z., Shah, R. U., Apte, J. S., Robinson, A. L., and Presto, A. A.: Restaurant Impacts on Outdoor Air Quality: Elevated Organic Aerosol Mass from Restaurant Cooking with Neighborhood-Scale Plume Extents, Environ. Sci. Technol., 52, 9285–9294, https://doi.org/10.1021/acs.est.8b02654, 2018.

3. Robinson, E. S., Shah, R. U., Messier, K., Gu, P., Li, H. Z., Apte, J. S., Robinson, A. L., and Presto, A. A.: Land-Use Regression Modeling of Source-Resolved Fine Particulate Matter Components from Mobile Sampling, Environ. Sci. Technol., 53, 8925–8937, https://doi.org/10.1021/acs.est.9b01897, 2019.

4. Shah, R. U., Robinson, E. S., Gu, P., Robinson, A. L., Apte, J. S., and Presto, A. A.: High-spatial-resolution mapping and source apportionment of aerosol composition in Oakland, California, using mobile aerosol mass spectrometry, Atmospheric Chemistry and Physics, 18, 16325–16344, https://doi.org/10.5194/acp-18-16325-2018, 2018.

---

## Author Comment (AC1)

**Mobile air quality monitoring and comparison to fixed monitoring sites for instrument performance assessment**

Andrew R. Whitehill, Melissa Lunden, Brian LaFranchi, Surender Kaushik, Paul A. Solomon

**Response to Reviewers:**

**NOTE: Reviewers comments are shown in blue text. Our responses are shown in black text.**

General Response:

We sincerely appreciate the thoughtful critique and suggestions provided by the 3 reviewers. Based on their feedback, we have made substantial structural and thematic changes to the manuscript in order to provide a more focused presentation of our results, their implications, and the strengths and weaknesses of our proposed approach for detecting systematic measurement bias in mobile monitors using "in-motion" mobile to stationary collocation. Instead of an extensive line by line accounting of changes, we provide the following high level summary of the changes made with more specifics provided in the direct responses to reviewers individual comments below.

- Title has been changed to: Mobile air quality monitoring and comparison to fixed monitoring sites for instrument performance assessment
- Abstract and Conclusions updated to reflect updated content in the results and discussion sections
- Introduction has been rewritten to be more focused and aligned with the following objectives of the paper:
    - Evaluate a scalable approach for identifying systematic measurement bias in mobile monitors
    - Explore the operational and performance trade-offs for using an in-motion mobile to stationary comparison in place if parked collocations
    - Quantify the magnitude of mobile measurement drift that can be quantified at different temporal aggregations
    - Explore what the data says about spatial heterogeneity (or the lack thereof) for the different pollutants considered and how this impacts these results
- Section 3 (Parked Collocations in Denver): We have de-emphasized the OLS linear regression results, focusing instead on ΔX as a measure of bias in Table 1
- Section 4 (Mobile Collocations in Denver):
    - We have added a discussion of how we are interpreting ΔX and $r^2$ and how these metrics relate to random spatial and temporal variability, persistent spatial variability, and systematic measurement bias.
    - We have added Figure 3 showing how ΔX and $r^2$ vary as a function of road type and distance from the site for each pollutant. The equivalent metrics from the parked collocations are included on these figures at distance = 0 in order to support the discussion of the comparison between parked and mobile collocations.
    - We have added Figure 4, which is analogous to Figure 3, but shows the impact of aggregating the 1 minute observations to 1 hour
- Section 5 (Mobile Collocations in CA):

- ○ We have added Figure 7, which is analogous to Figure 3, but for the CA data set (with no parked collocations included)
- ● Section 6 (New): Operationalization of mobile vs fixed site comparisons for ongoing instrument assessment
  - ○ Discusses a framework for how operational decisions could be made for efficiently incorporating mobile vs fixed site comparisons into drive planning to simultaneously meet mobile monitoring objectives
  - ○ Discusses the quantification of what temporal aggregation is required and what magnitude of drift (i.e. changes in systematic measurement bias over time) could be detected for a operationally feasible and scalable collection scenario
  - ○ Discusses the impact of using a minimum data completeness threshold for comparison of 1 Hz mobile data with hourly-averaged fixed site measurements
  - ○ Provides a case study of two NO2 sensors deployed in a mobile monitoring fleet and how systematic bias detection might work in practice
- ● Section 7 (New): Spatial Heterogeneity and Implications for Applicability to Additional Pollutants
  - ○ Discusses what the results in Figures 3 and 7 imply for spatial variability of the pollutants considered in this study
  - ○ Discusses the implications for using mobile collocations to identify systematic measurement bias for other pollutants of interest to the mobile monitoring community based on their expected degree of spatial heterogeneity

These substantial changes were prompted in large part by some common themes touched upon by all 3 reviewers. They each pointed out the relationship between pollutants having a high degree of spatial homogeneity in the atmosphere and the pollutants that have generally better correlations between mobile and stationary measurements. All 3 reviewers then go on to use this as a criticism of the study, suggesting that the high degree of correlation between mobile and stationary for O3, for example, suggests that hyperlocal mobile monitoring of this pollutant is not worthwhile. They also suggest that the poor correlation for pollutants, such as NO, means that the method for detecting systematic bias is not useful for the pollutants where mobile monitoring provides value. We realize that there are some shortcoming in our discussion of these concepts in the manuscript, and in our revised manuscript we have added a more in depth discussion of how spatial heterogeneity impacts the analysis and what the implications of our analysis are for spatial heterogeneity of the measured pollutants. The following is a direct response to this criticism from the reviewers and much of the language used here has been included throughout the text, as well as in a new section discussing the implications of our analysis for understanding spatial heterogeneity and the applicability for additional pollutants.

It is true that this study, in many ways, is an analysis of spatial heterogeneity of these 3 pollutants (NO2, NO, and O3) and how spatially representative a typical stationary measurement of these pollutants is. From this analysis, we come to what is likely not a very surprising conclusion that O3, being a secondary pollutant produced and removed via relatively slow photochemical processes is more spatially homogeneous, while NO, which is directly emitted and ultimately removed via relatively fast photochemical processes, is more heterogeneous. NO2, on the other hand, is in the middle of this spectrum, being somewhat more heterogeneous than O3 and less heterogeneous than NO, which is consistent with direct emissions, fast production (being the product of fast NO removal), and a relatively long lifetime to chemical removal.

Despite the important differences in spatial heterogeneity between these 3 pollutants, it is simply not accurate to use our results as an argument to suggest that there is little value in measuring O3 or NO2 on a mobile platform. To the contrary, our analysis does, in fact, show that all three pollutants have measurable and quantifiable spatial heterogeneity compared to the stationary site. This is particularly true when highway and major roads are included in the comparison, even for O3, which, of course, is fully expected due to the fast titration of O3 by NO. Figures 3 and 7 in the revised manuscript, for example, clearly show observable differences in the mobile measurements compared to stationary both as a function of distance from the site and as a function of road type. Our analysis intentionally removes highways specifically in order to reduce the spatial heterogeneity in the data set that results from increased density of direct emissions sources on highways, but there are likely some really interesting studies that one could undertake looking more closely at the interplay between the hyperlocal spatial patterns of these 3 closely related pollutants and the implications for effective emissions controls at hyperlocal scales. However, that is not the focus of this paper, nor is it our focus to justify such studies.

Instead, our primary aim is to quantify the amount of collocated data required over time to be able to compare the central tendency of the mobile and stationary monitor measurements in order to produce a reliable method for detecting differences and, further, to quantify the minimum differences in central tendency that can be reliability attributed to measurement bias and not natural spatial variability in the pollutants.

Additionally, we believe it is not an appropriate characterization to suggest that this method for detecting mobile measurement bias only works well for pollutants that, in the reviewers' minds, are more spatially homogeneous and therefore have less value in a mobile setting. While we acknowledge the reviewers' comments that the method works less well for pollutants that are more spatially heterogeneous, we would like to counter the argument that it does not work at all for these pollutants and thus has limited value. Taking NO, for example, if we were to attempt to derive a calibration correction for NO using a linear regression, we agree that it would not work well and acknowledge that some of our inclusion of regression statistics in the manuscript led to some confusion on this point. To clarify, it is not our intent to suggest using a linear model of hourly mobile and stationary measurements, in particular for NO, is a productive method to accurately identify or correct bias.

Rather, our analysis shows that by collecting enough mobile collocation data for comparison, we can reduce random differences due to the combination of random (gaussian) measurement error and spatial variability in order to determine the central tendency of those differences to within some uncertainty bounds. There is an important conceptual framework around spatial variability that is key for understanding why it is possible to use the aggregated mobile-to-stationary differences to measure systematic measurement bias even in cases like NO where the linear regression of hourly measurements is so poor. The time scale for spatial variability is critical here. For any given hourly snapshot, the measured mobile value may not be a good predictor of the stationary concentration for that same hour, as our analysis suggests for NO. However, when aggregated over a longer period of time, the mobile vs stationary differences are likely to be much smaller as random spatial variability is averaged out as the vehicle samples more of the existing variability in atmospheric concentrations and concentrations in the region.

The entire field of hyperlocal air quality monitoring is predicated on the fundamental principle that repeat samples or passes collected in the same location over time will result in an aggregated value that

will approach and converge upon a meaningful central tendency for that location. The reason why it is important to collect repeat samples is that temporal variability is typically much larger than spatial variability, but if enough samples are collected, the central tendency can be accurately observed for each location and the true spatial variability across a neighborhood, city, or region can be discerned.  There is a direct analogy between this concept and the method we are describing in this manuscript.

Towards this end, we present what we think is a very useful finding, that, even for NO, the uncertainty in that central tendency of mobile to stationary differences decreases with an increasing number of comparison data points aggregated and converges on a relatively constant value after about 40 distinct hours of comparisons. The important quantitative result is that with a 40 hour running median window, we can attribute differences greater than ±8 ppb NO between mobile and stationary to measurement bias with high confidence.

We do not consider other pollutants such as BC or UFP in this study, which, as reviewer 2 suggests typically have a higher degree of spatial heterogeneity similar to NO; however, we presume that a similar relationship would exist between $\Delta X$ and the median window size as that shown for NO, NO2, and O3 in Figure 8 (in the revised manuscript). The differences would most likely manifest in the optimal window size and the width between upper and lower percentile values at which the curves converge. If one was interested in applying this method to other pollutants, we have provided a blueprint for doing so, and we believe that it may be feasible for many pollutants depending on the magnitude of expected sensor drift, the magnitude of temporal and spatial variability of that pollutant in a given study area, and the specific monitoring objectives.

**Reviewer 1 Comments and Responses**

In their manuscript "Mobile Air Quality Monitoring and Comparison to Fixed Monitoring Sites for Quality Assurance", Whitehill and coauthors present comparison measurements of ozone, NO, NO2, and Ox (O3 + NO2), measured on-board mobile platforms with those measured in stationary air quality network stations. The mobile platform measurements were performed both, during parking near the network stations and during mobile measurements in the vicinity of the network stations.

Linear regression analysis of the data from the mobile and the stationary platforms showed good agreement, albeit with small systematic negative biases for O3 and NO2, and large coefficients of determination for ozone, NO2 and Ox for both data sets from both platforms. NO data, however, did show much poorer agreement and correlation, compared to the other variables.

Generally, agreement between the mobile platform data and the network station data was better when the mobile platform data were collected closer to the network station. Nevertheless, for the above mentioned variables reasonable agreement between the data was found for distances up to 10 km from the network station.

These results were used to develop a method to quality assure mobile measurement data from long-term field measurements by using data acquired within a certain distance to an air quality network station and using median values over 40 hours of data.

The manuscript is well structured and clearly written. The results show that for some pollutants, comparison of mobile data with data from an air quality network station in the vicinity might be used to determine calibration drifts or malfunctions of mobile instruments under certain conditions. For this purpose, the manuscript will be useful for applications where mobile measurements were conducted over extended times within a certain area, which also contains a fixed monitoring site. However, the poor comparison results for NO, which is a pollutant, which is dominated by local sources, shows that such a quality assurance approach can only be applied for pollutants which have a very homogeneous spatial distribution over an area of several km extension. Strong spatial variability of pollutant concentrations will make such an approach impossible. This limitation will likely not only apply for NO (which was measured in the study), but also for particle-related variables like BC or particle number concentration, which also show a large spatial inhomogeneity.

It would be desirable, if the authors would more critically assess their approach with respect to such limitations also in the general sections of their manuscript like the abstract and the conclusions section. This critical assessment should also include the fact that regionally homogeneous distribution of pollutants – which is necessary for the suggested in-field quality assurance method – also likely makes it less necessary to map out pollutant distribution with a narrow-gridded driving pattern, which takes a lot of effort to perform. The balance between mapping out pollutants with sufficient spatial resolution and having the possibility of using "remote" stationary measurements for quality assurance – potentially by applying sufficient temporal averaging – should be discussed.

See our discussion in our "General Response" above. Significant modifications were made to the manuscript to address these issues, including additional discussion of other pollutants.

I recommend publication in AMT after these and several other minor issues have been addressed, as detailed below.

Detailed comments:

Section 2 – Overview of Methods:

According to previous publications and the Aclima website, also black carbon and particle number concentration was measured on Aclima vehicles. It is not clear, whether this was also the case during the measurements used for this study. Since especially these particle-related variables probably have a very large spatial (and temporal) variability, it would be very relevant to include such data in this assessment of comparability of mobile and stationary measurements.

Added the following text to Section 2 to address this:

*Additional measurements, including black carbon (BC), size-fractionated particle number counts (PN), and other species were also measured during these campaigns but are not discussed in this manuscript. For this manuscript, we focus on measurements that had equivalent mobile and fixed reference site measurements for both studies. A critical part of this work is our comparison of parked and mobile collocations during the 2014 Denver study, for which we had one-minute averaged reference site data for O₃, NO₂, and NO but not the other measured species.*

Section 3.1 – Mobile platforms parked at reference site - methods:

How large was the distance between the mobile measurement platform and the stationary measurement setup inlets? Which were the altitudes above ground level for both inlets? Further below (caption of Figure 2) it is stated that the distance was between 80 and 145 m and between 10 and 85 m for the two sites. Why did the distances vary so much? For both sites, the largest distances of the parking locations to the measurement sites are larger than the distance to the closest roads. Why was the vehicle not parked directly at the site for comparison? How large is the influence of the distance on the comparability of the results?

We do not have information on the precise altitudes above ground level for the stationary monitors. The mobile platform sampled from a few inches above the top of the front windshield (so ~1-2 m a.g.l.). Given the large horizontal distances between stationary and the parked locations, the differences in altitude are very likely insignificant. We have added the following text to Section 3.1 to explain why the car-to-site distances were so large:

*The drivers were instructed to park as close as possible to the site but were required to park in an available public parking space on a public street. These restrictions limited how close each car could get to the reference site for each collocation and resulted in individual parked collocation locations ranging in distance from the regulatory stations, as shown in Figure 1.*

Section 3.2 – Mobile platforms parked at reference site - Results and Discussion; Line 188-193:

Can the typical differences between the car measurements and the fixed site measurements be explained by the sampling situation? Are these differences arbitrary differences or can they be explained by the sampling environment (e.g. traffic), meteorological conditions (e.g. wind direction) or other external influences? Do these differences and the variability of these differences (i.e. the r2 values) between the car and fixed site measurements reflect the spatial inhomogeneity of the respective pollutant concentrations? How are they related? How do they depend on the distance between both sampling locations? As a consequence, can (mobile) measurements of spatial inhomogeneity of pollutants be used to estimate how well such a measurement comparison (or the quality assurance approach presented further below) will work for a certain pollutant?

We added significant discussion to the revised manuscript about the relationship between $r^2$ and the spatial heterogeneity of different pollutants. See Sections 4.2 and 7 of the revised manuscript.

Section 4.1 – Mobile platforms driving around a fixed reference site in Denver – Methods:

Line 198: Impact by emission plumes from mobile sources is not only a problem for parked vehicles, but much more for driving vehicles (i.e. mobile measurements), which are often surrounded by other vehicles, driving on the same road.

It is our view (supported by our previous publication, Whitehill et al. 2020) that the impact of emission plumes is less for mobile than stationary comparisons unless ideal (off-road) conditions are used for the stationary comparisons. We added the following text to Section 3.2 to address this:

Although it is possible to impose strict collocation criteria for parked collocations that would limit the influence of local emissions, the operational constraints during large-scale mobile monitoring campaigns often necessitate the use of publicly-accessible sites for frequent collocations. Since most scalable parked collocation solutions are likely to be affected by traffic emissions, expanding to allow the use of additional data while driving in the vicinity of the fixed reference station should be explored as a viable alternative. Mobile collocations have the added advantages that spatial biases are averaged out by the motion of the mobile platform through space, effectively allowing each mobile datapoint to sample a larger (and, by extension, more representative) amount of air in the same sampling duration (e.g., Whitehill et al., 2020)

Line 212: Can "lower traffic" be somehow quantified? An "empty" highway affects the measurements probably less than a congestion in front of a traffic light or at an intersection on a residential road.

In this manuscript, we used OSM road type as a general proxy for traffic. In the revised manuscript, we added text to Section 4.1 to state that explicitly:

Based on our results from Section 3, we believe that the measurements made on Residential roads will generally have lower traffic, and thus stronger agreement with measurements at most fixed reference sites. While traveling on high traffic roads (such as highways), the cars are more likely to be impacted by direct emission plumes. In effect, we are assuming that the OSM road classifications is a general proxy for on-road traffic volume.

Table 2: This is rather detailed information which could be shifted into the supplement.

This table was moved to the supplement as suggested.

Figure 3: In this figure correlations between 1-min averaged ozone concentrations measured at a fixed site and during mobile measurements at distances up to 10 km are shown. For distances larger than may be 100 m, it is clear that different air parcels are compared in every data point. Therefore, these correlation plots are not a comparison of the performance of the measurement instruments in the car and on the fixed site, but an investigation of spatial homogeneity of the ozone concentrations. Since ozone is rather homogeneously distributed, such a comparison can be used for quality assurance of the instrument (at the same time it is questionable, why in such a case small-meshed mobile measurements should be made). For other pollutants like NO (but likely also BC or PNC), the correlations are (or would likely be) much weaker and consequently the quality assurance approach would not work. All this should be treated and critically discussed in such a paper which focuses on the possibility to use such measurements as quality assurance approach (not only in a short sentence, which explains why NO shows much weaker correlations – line 262/263).

In the context of this work, we consider correlations ($r^2$) to be more of an indicator of random spatio-temporaltemporal and spatial heterogeneity of pollutants than a measure of real instrument / measurement bias. See our general response for more detail. We also have added the following in the revised manuscript (in Section 4.2):

In general, the mean and median bias values can reflect both measurement bias as well as persistent spatial differences. The $r^2$ values generally reflect the random variability between the mobile and stationary measurements that results from a combination of measurement precision as well as true spatial variability. Both the bias values and the $r^2$ values for the in-motion observations reflect additional sources of spatial variability that need to be considered as compared to the parked collocations. This is due to the wider range of distances (up to 5 km), varying road types and associated traffic patterns, and potentially different spatial distributions of non-mobile sources in the wider areas covered. As discussed in Section 3, $r^2$ values vary depending on the pollutant measured, and can be quite low even for parked collocations. As a result, we conclude that $r^2$ would not be a good indicator of instrument performance (i.e., precision error) and that using a parametric linear model to attribute gain and offset instrument biases separately is not possible, particularly for NO and $NO_2$. Therefore, the inclusion of $r^2$ in this section is primarily as context for understanding random spatial variability in the comparisons.

In addition, I think all these correlation plots could go into the supplement and rather a plot that shows r2 (and slope in another panel) versus distance for the various road types should be shown here. The same is true for Figure 4.

In the revised manuscript, we have added new figures (Figures 3, 4, and 7) that show $r^2$ as a function of distance for various road types. We move all figures showing slopes and intercepts to the supplement since our focus is on central tendency metrics not least squares regression statistics.

Line 268-270: I agree that it is important to exclude local emission plumes from the averages. I wonder why the highly time-resolved data were not used to exclude such plumes before averaging, like it was shown in this journal earlier for mobile measurements (e.g. Drewnick et al., AMT 2012).

The use of medians in our temporal aggregation essentially achieves the same effect as other more sophisticated methods for discriminating between background and local emissions plumes (see, e.g., Brantley et al. 2014). While there could be some value in exploring some of these additional methods in future refinements of the approach, it is beyond the scope of this study.

Line 372-374: To me it seems rather doubtful that a few seconds of data can reasonably represent a one-hour average. How would the comparison (mobile/stationary) change or improve, if a minimum data coverage would be introduced?

Added a discussion in Section 6.1 of the impact of using a "minimum data threshold" on the comparison. Figure 8 in the revised manuscript shows the difference in the range of running median hourly ΔX values as a function of minimum data threshold for three values (no threshold, 5 minutes, and 10 minutes).

Section 5.3 – Using driving data for ongoing performance evaluation:

If I understand this performance evaluation approach correctly, it assumes a constant drift of the calibration of the mobile instruments over longer time intervals or a malfunction which causes a change in the calibration or the response of these instruments, in order to be able to detect the bias. What about temperature-related drifts in instrument calibrations, which would possibly occur repeatedly over the course of a day? Would such biases be detected by this approach?

For the approach described, the type of bias that is most readily detectable would be long term drift over the time it takes to compile the 40 collocation events required for the running median statistic. It could be possible to detect diurnal changes in bias (i.e. from temperature related effects) if enough collocation events could be compiled over time across enough different hours of the day. There may, however, be some additional complexity with this approach as there could be cyclical systematic spatial differences across diurnal timescales that would need to be well characterized. Long term measurement drift in a single direction would be a much more distinctive pattern compared to any long term changes in systematic spatial differences. While looking at diurnal, day of week, or seasonal patterns would be an interesting analysis, it is beyond the scope of this study.

Figure 8 – caption: shouldn't it be "running median" instead of "running mean"?

Corrected figure caption in the revised manuscript (Figure 9).

Line 450-455: How do you know that these DeltaX-values actually reflect changes in the calibrations of the mobile instruments and consequently could be used to correct for those? Couldn't there be other reasons for these differences, like local sources or an issue with the stationary instruments?

New text in Section 4.2 (and elsewhere) discusses how the $\Delta X$ values are impacted by both persistent spatial variability as well as measurement bias. We also agree and acknowledge that measurement bias in the case of two sufficiently calibrated instruments primarily reflects inter-lab differences. We note this now in Section 4.2, referring to the "measurement bias between mobile and stationary monitors", as well as Section 5.2, where we discuss possible differences in inter-lab comparability between the Denver and CA data sets.

**Reviewer 2 Comments and Responses**

This review is for the above manuscript submitted for publication in Atmospheric Measurement Techniques. The manuscript partially develops and proposes implementation of a new quality assurance procedure to evaluate changes in instrument performance in mobile monitoring of air quality. To do that, the authors use high-temporal resolution (O ~ 1s) mobile-monitoring data collected using regulatory-graded instruments from two campaigns conducted in different regions for very different lengths of time for three pollutants, O3, NO2, and NO. The authors then compare stationary referencing of this data during collocations with the regulatory monitor to the referencing of "vehicle-inmotion" concentrations with regulatory monitoring data (based on distance and road type from the regulatory monitor) for one site, and find similar performance evaluations for a regionally distributed pollutant O3 (r2 > 0.9), moderate performance for NO2 (r2 > 0.5), and poor performance for a primary pollutant, NO (r2 < 0.2) in their new approach. For the second site, the authors do not conduct stationary referencing and only perform the latter "vehicle-in-motion" referencing to the regulatory monitor to estimate optimal temporal "running windows" to identify instrument issues. They calculate that for a 3 km spatial window, a temporal running window of 40 hours for data would allow detection of a systematic measurement drift or sudden instrument or sensor malfunction over the time scale of 7-9 days. While the authors motivate this work to identify and address systematic measurement drift or malfunction, the same is are not demonstrated in their results nor do they show the implementation of this method on any dataset. Additionally, in its current format, the manuscript uses linear models even when they do not seem applicable, especially for NO and NO2. Finally, the deficient analysis of high concentration plumes further limits the usefulness of the proposed method. I recommend that the authors significantly revise and resubmit this manuscript for further consideration.

We revised the manuscript to de-emphasize the linear models and further reinforce that linear regressions are a poor choice for performing these comparisons. In addition, we clarify in Section 4.2 that we consider $r^2$ to be an indicator of random spatial heterogeneity rather than "performance" and prefer alternative methods for assessing performance.

1. Deficient motivation and lacking significant findings: The authors begin by motivating this work as to "provide an important tool for ongoing quality assurance during mobile measurement campaigns" [Line 51]. They believe that "through ongoing comparisons of fixed reference site and mobile measurements, it may be possible to identify instrument drift over time or changes in instrument performance that could indicate a malfunction" [Lines 47-49]. The other motivation the authors suggest is that ongoing "mobile-versus-fixed-site comparisons are more scalable than frequent site-by-side parked collocations", "which is particularly important during sustained, multi-vehicle (and fleet-based) mobile monitoring campaigns" [Lines 50, 57, 58]. However, in their findings, the authors conclude and I agree that this method "is not an absolute method for calibration or instrument verification, as a direct collocated comparison with reference monitors is" [Line 458]. In the two campaigns the authors conducted, they performed daily checks with zero and span gases and in one campaign, conducted direct collocation comparisons. These standard approaches provide lab-grade confidence in the measurements including with regards to instrument drift and malfunction and no replacement for them has been identified in this work. While authors do find that regional pollutants are correlated strongly in "mobile-versus-fixed-site comparisons", these pollutants are expected to exhibit regional homogeneity and this result is not a significant contribution of this work. Frequently, measurements dominated by secondary pollutants are referenced to nearest (but farlocated) reference monitors, both in stationary and mobile monitoring.

What the authors in fact demonstrate is the weakness of the "mobile-versus-fixed-site comparisons" for pollutants with high spatial variability such as NO. While 15-16 stationary collocations only 20 mins each conducted over 2-3 weeks in the Denver campaign to calibrate against regulatory monitors yield mean r2 of 0.4, this comparison performance drops precipitously to r2 <= 0.2 in hourly averaged "mobileversus-fixed-site comparisons" (Table 3). This is not surprising, since "spatial coverage from mobile monitoring reveals patterns missed by the fixed-site network", especially for primary pollutants (Chambliss et al., 2020).

Our opinion is that it is an inaccurate characterization to say that there is little value in monitoring O3 or NO2 on a mobile platform. There is significant spatial heterogeneity at fine scales in all the measured pollutants, which is even visible in the central tendency aggregations (see, e.g., Residential vs Highway roads in Figure 3 in the revised manuscript). Also see our general response above.

Anyway, scaling their standard stationary collocations up to a year (approximately equivalent to the length of the California campaign, also presented here) totals to about 60 hours. In this work, the authors propose sampling in their newly developed approach within a crude spatial scale of 3 km for about 40 rolling hours to identify instrument drift/malfunction. Clearly, there is little advantage to switching to this new approach given the drop of ~50% in the measure of performance in the "mobile-versus-fixed-site comparison" in the only primary pollutant monitored compared to the standard "stationary collocation". In short, while the authors argue that stationary collocations "ensures comparability only at that specific location and only under the specific atmospheric conditions over which the collocation occurred" (Lines 37-38), they have demonstrated that stationary collocations perform significantly better than their proposed method.

In the revised draft, we have presented the results of the comparison between stationary collocations and the in motion mobile vs stationary collocations in a way that we feel makes our point in a more compelling way than was conveyed in Table 3 of the original manuscript. Figure 3 has now been included showing how 3 critical metrics (mean $\Delta X$, median $\Delta X$, and $r^2$) vary by road type and with distance from one of the Denver sites (La Casa). We have also added an additional description in Section 4.2 discussing how to interpret these metrics, noting that $\Delta X$ is an indication of both persistent spatial differences as well as any existing measurement bias and that $r^2$ is an indication of random spatial, temporal, and measurement variability. The equivalent metrics from the stationary collocations are included as single black circles on the same figure, located at distance=0 on the x-axis.

We note that while there may be a decrease in the $r^2$ from stationary to mobile collocations, primarily for NO (but negligible for $O_3$ and $NO_2$), this decrease is relatively inconsequential to the method for assessing measurement bias, which is indicated by $\Delta X$. Since we are not attempting to adjust instrument gain and offset using a slope and intercept from an OLS (or any other linear regression approach), the $r^2$ really only impacts the amount of temporal aggregation required to reduce the uncertainty on $\Delta X$ and, ultimately, the magnitude of that uncertainty. As we show in Figure 9 in the revised manuscript (and Figure 8 in the original manuscript) all pollutants show stable $\Delta X$ estimates with a median window size of approximately 40 hours of collocation, despite there being widely varying $r^2$ for each pollutant.

We argue, from Figure 3 (in the revised manuscript), that there is negligible difference in the assessment of $\Delta X$ when using mobile to stationary collocations at close distances (~500 m) on residential roads and that the benefit in efficiency gained of keeping the vehicle moving and not having to find appropriate parking near the site is well worth any minor trade offs in performance. In Section 5.2 we further explore

these trade-offs of using a wide buffer distance (of 3 km) with the more extensive California data. This represents a substantial increase in collection efficiency with a very clear added benefit over an equivalent amount of time spent parked near a stationary site.

2. Use of linear models: As Figures 1 and 4 demonstrate, a linear model seems insufficient to compare measurements of NO and NO2. There is a clear baseline effect, where only a small fraction of variance in concentrations can be explained by variations in the reference site. The extensive dependence on presentation using linear models in the manuscript for the Denver phase further weakens this manuscript. I suggest the authors' reconsider the presentation approach as well as the feasibility of a "mobile-versus-fixed-site comparison" given visible baselines that are developed primarily based on comparison with the Denver "stationary collocations".

The emphasis on the OLS statistics of the comparisons in the original manuscript was causing undue confusion and has been largely removed. Scatter plots (and the associated OLS statistics) are commonly used and often useful visualizations and descriptions of collocation data, and therefore have been retained, but moved to the supplemental information. We use $r^2$ throughout the text as an important indicator of random spatial and temporal variability as well as random measurement variability (i.e. precision error) and have included a detailed discussion of how it should be interpreted (primarily in Section 4.2).

3. Studying outlier plume events: Hyperlocal monitoring has a lot of value given its ability to map pollutant exposures, especially plumes of primary pollutants with high spatial variability. However, prior work suggests that primary pollutant spatial patterns can be predicted well using land-use regressions (Robinson et al., 2019). In contrast, regionally distributed pollutants are not well represented by such regressions but are relatively spatially homogeneous and can be estimated using regional monitoring (Shah et al., 2018). Given these well-established priors, the baseline effect in (2) above, the high concentration plumes or "outliers" in primary emissions should be studied separately. Similar mobile monitoring work has been done previously and could be referenced (Robinson et al., 2018).

This work is focused on instrument performance assessment. Studying specific outlier plume events is beyond the scope of the present manuscript.

1. Chambliss, S. E., Preble, C. V., Caubel, J. J., Cados, T., Messier, K. P., Alvarez, R. A., LaFranchi, B., Lunden, M., Marshall, J. D., Szpiro, A. A., Kirchstetter, T. W., and Apte, J. S.: Comparison of Mobile and Fixed-Site Black Carbon Measurements for High-Resolution Urban Pollution Mapping, Environ. Sci. Technol., 54, 7848–7857, https://doi.org/10.1021/acs.est.0c01409, 2020.

2. Robinson, E. S., Gu, P., Ye, Q., Li, H. Z., Shah, R. U., Apte, J. S., Robinson, A. L., and Presto, A. A.: Restaurant Impacts on Outdoor Air Quality: Elevated Organic Aerosol Mass from Restaurant Cooking with Neighborhood-Scale Plume Extents, Environ. Sci. Technol., 52, 9285–9294, https://doi.org/10.1021/acs.est.8b02654, 2018.

3. Robinson, E. S., Shah, R. U., Messier, K., Gu, P., Li, H. Z., Apte, J. S., Robinson, A. L., and Presto, A. A.: Land-Use Regression Modeling of Source-Resolved Fine Particulate Matter Components from Mobile Sampling, Environ. Sci. Technol., 53, 8925–8937, https://doi.org/10.1021/acs.est.9b01897, 2019.

4. Shah, R. U., Robinson, E. S., Gu, P., Robinson, A. L., Apte, J. S., and Presto, A. A.: High-spatial resolution mapping and source apportionment of aerosol composition in Oakland, California, using mobile aerosol

mass spectrometry, Atmospheric Chemistry and Physics, 18, 16325–16344, https://doi.org/10.5194/acp-18-16325-2018, 2018.

**Reviewer 3 Comments and Responses**

Paper summary, in my words:

This paper presents an assessment of measurements made from mobile platforms against fixed-site regulatory measurements as a comparison benchmark. The authors describe approaching the above in two different ways: parked collocation between mobile and fixed-site, and in-motion comparison between mobile and fixed-site monitors across different buffer sizes, road classifications, and averaging intervals. The stated broader vision of the paper is to provide a framework for identifying and addressing instrumental problems (e.g. drift) in a certain type of dataset (multiple mobile monitors doing highly-routinized driving over long durations).

When comparing stationary mobile monitoring data vs the fixed-site monitoring data in Denver, the authors quantify the agreement between the Aclima cars and each fixed-site, concluding that the stationary mobile-monitor does well for O3, fairly well for Ox (though not as well as hypothesized), ok for NO2, and poorly for NO. The conclusions are based on r2 values and slopes from OLS as metrics. The authors discuss the results in the context of regional vs. primary local pollution.

Our goal was to focus on central tendency metrics (e.g., ΔX) as an indicator of bias. We largely de-emphasized OLS regression statistics in the revised manuscript to reduce confusion on this aspect.

The authors then use in-motion mobile monitoring data to assess by-pollutant variability across road category and spatial aggregation buffer size in two different locations (Denver and California). The stated difference in these analyses is that the Denver reference monitors had higher time resolution (1-minute) compared to those in California (1-hour), but had less temporal and seasonal coverage.

The general conclusion from the Denver in-motion analysis is that mobile data works similarly to data collected while parked when comparing against a fixed-site. Measurements collected closer to the fixed-site monitor tended to be better than data collected further afield. The authors allude to a tradeoff between including more measurements (which tend to make the comparisons more robust, all else equal) and including measurements made further away (which tend to be weaker comparisons due to spatial variability).

The authors then address the above by considering how much data is needed, temporally-speaking, as they do this analysis at a single fixed buffer size, to make an assessment of whether or not a mobile monitor is agreeing with a fixed-site monitor. They conclude that for all pollutants, 40 hours of data is what is required.

Overall analysis of paper:

I think it is important for the field to continue developing best practices in mobile monitoring to ensure data quality and distill meaning. This manuscript does work in that regard that is novel enough to warrant publication, after significant revision. There are only likely to be more of these big-scale mobile monitoring efforts, and moving towards transparent, algorithmic QA practices is a good thing.

Unfortunately, the analysis here does not offer much depth or applicability for the key thing that mobile monitoring has the power to uncover—spatial variability in pollutant concentration for primary emitted

pollutant, such as NO, which performs poorly in all comparisons outlined by the authors. Presumably things like black carbon, particle number, methane/ethane, many types of VOCs, and certain primary organic aerosol constituents would all perform more like NO than O3. The authors do not offer much in the way of explanation of what their results mean for (arguably) the most important pollutant that they measured, given the "strong suit" of what mobile monitoring can address. This seems like a real hole in the manuscript that I think needs to be (much) better addressed and discussed.

Relatedly, the results that do compare most favorably are for pollutants that are much less spatially variable. While it isn't a bad thing to perform this analysis on these pollutants (e.g. O3), and in fact serves as a good 'base case check,' these are not the type of pollutants that one goes to the great lengths of having a fleet of mobile monitors for. Again, I think this is a conceptual hole in the manuscript.

See our General Response above. Significant modifications have been made to the revised manuscript to further explain and clarify our goals in these analyses and our interpretation of the results.

Lastly, I have a lot of more detailed criticisms about figures, tables, etc. that make this paper not publishable in its current form. I suggest both a systematic re-assessment of the results in this paper in the context of primary pollutant spatial variability and addressing the detailed "minor" issues below, followed by a re-review, before publication.

Big(ger) picture stuff

-There are a lot of pollutants, including ones of interest to many (PN, BC) that are mentioned as being measured, but are nowhere in the results. Why? Please address/discuss.

Added the following text to Section 2 to address this:

*Additional measurements, including black carbon (BC), size-fractionated particle number counts (PN), and other species were also measured during these campaigns but are not discussed in this manuscript. For this manuscript, we focus on measurements that had equivalent mobile and fixed reference site measurements for both studies. A critical part of this work is our comparison of parked and mobile collocations during the 2014 Denver study, for which we had one-minute averaged reference site data for $O_3$, $NO_2$, and NO but not the other measured species.*

-Section 3 - the lack of agreement between NO from the car and the fixed-site must (assuming both instruments work/were calibrated, as they are stated to) be due to spatial variability between the parking spot and the monitor. The parking location is mentioned as a range of distances, and clearly where the car is parked (and/or which way winds were coming, etc.) should explain the disagreement. This should be addressed in some detail.

Added the following text to Section 3.2 to address the source of the NO discrepancies:

From the combined time series of all collocations (Figures S1 and S2), short-term peaks in NO and $NO_2$ are present in the mobile platform measurements but not the fixed reference site measurements. This reflects the impact of emission plumes from local traffic. The traffic influences are particularly noticeable at the CAMP site, reflecting its location at a major intersection.

Additionally, given the reliance on the difference measurements in Section 5, I would expect more of a discussion of the difference measurements in the last part of Section 3. It appears that we are looking at an ozone monitor that has a systematic offset from the fixed-site?

In Section 3.2 of the revised manuscript, we acknowledge the possibility for a real systematic offset:

Both the 25$^{th}$ and 75$^{th}$ percentiles for $\Delta O_3$ were negative as well, suggesting a real differences in the ozone measurements between the mobile platform and the fixed reference site.

-Section 4: Suggestions for improving the scatterplots are presented below. But more generally, I think that a figure that synthesizes the findings beyond the many scatterplots should be present, e.g. R2 vs. buffer size, for all pollutants. As mentioned above, I think much more emphasis needs to be spent on what these results mean for pollutants that are expected to be spatially heterogenous. Are we really looking at quality-checking the analyzer in a mobile lab compared to the ground-truth fixed-site, or are we simply comparing different air masses with different concentrations? This paper is presented as a means of quality-checking baseline instrument performance. But a lot of the pollutant comparisons are really assessments of their spatial variability, which need not have anything to do with instrument performance. Throughout the manuscript these ideas seem conflated.

Figures 3, 4, and 7 now show $r^2$ vs buffer size for different road classes. All scatterplots (except for Figure 2) were moved to the supplement. We significantly increased our discussion of the implications of our results for spatial heterogeneity, including the added Section 7.

Section 5: a lot of the value of Section 5 compared to Section 4 seems to be: how does the above approach work with the more-common, longer-duration (hourly) fixed-site measurements? And the difference in the temporal coverage between the two datasets (Denver and California) is what allows the authors to pursue this question. However, the authors then average all of their measurements to hourly, but say that even very brief intervals (1s) can be used for these hourly averages. To me, this does not seem justified, especially for pollutants with spatial and/or temporal variability (which again are largely the kind of interest to the mobile monitoring community). Moreover, I would think that shorter-duration mobile monitoring averages would be more scattered than longer-duration averages, all else equal. Is this so? How does this vary by pollutant? The authors should justify this choice, I think.

Added discussion in Section 6 (of the revised manuscript) about the impact of setting "minimum data thresholds" on the one-hour aggregations. We also revised Figure 8 to show the impact of using three different minimum data cutoff values when performing our analyses.

Also (from Section 5), Figure 8 seems to be a major distilled result that the authors are driving towards with all of the previous analyses. However, some of the fundamental aspects of this Figure were hard for me to understand. Why is the "range of median differences" the important quantity here? And should this quantity be normalized in some way, as Figure 8 makes it seem like NO agrees more than O3 for a window see of 20 hours or less?

We significantly modified Figure 8 to show the actual range (minimum and maximum) for each pollutant, and separated each pollutant into a different frame for ease of visualization. We hope that the revised figure is easier to understand than the initial figure. We also added the following text (in Section 6.1) to make sure the reader understands why the range of median differences is an important quantity:

The range between the upper and lower traces in Figure 8 provides an estimate of the uncertainty in median $\Delta X$ due to random spatial variability, and thus a measure of the magnitude of change in systematic measurement bias that we can expect to be observable by this approach.

Things to change/reconsider in tables and figures:

-Table S1 (and S2): I don't think anyone will particularly care about Start Time and End Time. However, it would be useful for the reader to be able to see how many minute-avg. points each stationary sampling period yielded—please add this.

Change to Start Time (useful to determine the time of day) and Duration. Also added supplemental figures showing time series of each stationary collocation.

-Figure S1: What values are the whiskers signifying? This should be stated somewhere. What do the points (outliers? Defined how?) signify? This should also be stated. It seems a bit hard to believe, looking at the NO plots from figure 1 that there would only be one outlier point at each site in the deltaNO quantity—is something off here? Also, this should be in the main text and not the SI, in my opinion. Also, I am surprised that the results in this figure are not connected to the analysis at the end, or discussed in that context—clearly there is a systematic offset between the mobile monitor and each fixed-site measurement of ozone, which warrants some discussion.

This figure was removed and the relevant statistics are included in the main text in Table 1 instead.

-Figure 1: This comment will apply to all of the scatterplots: For figure 1 you are showing the one:one line, but not showing the OLS results. For many of the other similar plots you show OLS results but not the one:one line. Given the variety of scales used (between pollutants, spatial aggregations, etc.), I strongly recommend adding both of these (in different colors) to each plot, as quick visual reference points for the reader.

We moved all OLS regression plots to the supplement to de-emphasize this analysis in the main text. We added both 1:1 and OLS regression lines to scatterplots in the supplement.

-Figure 2: This figure is meant to establish the spatial context of the two sites, including the length scale and position of parking relative to the stationary monitor. I don't think this figure accomplishes these goals. There should be distance scale bars. Given the emphasis on distance-from metrics in later sections, it would also be helpful to see how these sites fit into any larger land-use context. Also, given the large range of car-to-site distances, some kind of areal shading makes more sense than a single point marking parking. Also, this should come before Figure 1. Other things would be useful to show here too (which were not discussed in the text at all, and possibly should be): wind-rose insets (Seems like it would be quite important potentially for the La Casa site, though again the reader has no way of knowing what is west of the monitoring location (assuming that these maps are oriented with north being "up")), and north arrows.

We revised and improved all the maps in the manuscript, including adding shading to indicate parking areas and adding wind roses and scale bars.

-Table 2: While this information is important, this seems less relevant for the main text than pretty much everything currently in the supplement. I suggest moving this to the SI.

This table was moved to the SI as suggested.

-Figure 2 (and 3): again, please add one:one lines, and again this is too low of resolution to be useful. I would also suggest some kind of different plotting style given that it is too difficult for the reader to see

the data for e.g. 100m-Major. A bigger picture comment mentioned above, but I will re-emphasize it here: that ozone (a regional pollutant not expected to have much spatial variability) behaves well is not surprising. I'm not saying it isnt worth presenting this result, but it makes no sense to me to present O3 in the main text and NO in the SI, given that NO is spatially variable, and hence of interest to this whole endeavor in a way that O3 really isnt.

We moved all of these figures to the supplement and added one:one lines to all of these figures.

-Table 3: Why would you omit 100m, 3000m, etc. here? You make a point to display the plots in the previous figure, but then only show a subset of the regression results. Whatever goes in Figure 2 should go in Table 3 (or vice versa).

We removed this table and instead show the relevant information (stationary versus mobile comparisons) in Figure 3.

-Figure 4: shouldn't N for the individual non-highway categories (e.g. major, residential) sum to the N shown in the bottom row "Non-Highway" category? I am noticing that for the 100m column they do not, on this figure.

Added text to the caption to explain why the N for non-highway should be less than the sum of N for the individual categories.

-Figure 5: This map also does not effectively communicate what the authors presumably intend, and it needs significant work. I am guessing (though it is not explicit) that the colored roads are where the Aclima cars drive? Or are these a subset of the roads drive, but the sections of which that are within the largest buffer size? Its unclear. There should be scale bars and a north arrow here. At this level of zoom it's impossible to tell residential from major roads. I would consider substantially re-working this figure to best communicate the relevant ideas—where the reference monitors are (and probably label them on the map), how the buffers around each compare to the spatial extent of the domains (so perhaps just draw some rings). I would let the inset maps "do the talking" as far as the road categorization goes, because it's very difficult to get a sense for that at this level of zoom. There are other minor, but still important aspects that need work here, such as the marker size for the reference monitors—one of the most important pieces of info you are trying to convey—is a very small fraction of the font size of the legend.

We revised and modified all of the maps in the main text following these suggestions.

-Figure 6: I don't understand what the authors are trying to convey with this figure, exactly. And why just show Livermore, but not the other monitoring sites that are mentioned? I suggest showing each monitoring station I think some text is needed to very clearly outline the message of the figure ("As you can see from Figure 6, the driving routes are concentrated around the fixed site. Also, you can see how each of the three monitors (West Oakland, Livermore, San Francisco) compare to each other in terms of vehicle-related land use in the vicinity of the monitor as well as where we drove). These maps also should have scale bars, north arrows, and buffers indicating the relevant buffer sizes used in the analysis. All of that would make this figure convey much more spatial context than it currently does.

We show all three of the main sites in the revised figure (Figure 6), as well as showing scale bars, north arrows, etc.

-Figure 8: The figure caption says mean, while the y-axis says median. I also suggest adding as horizontal lines the bias from the zero/span checks, for comparison (as is done in the text).

Fixed caption in revised manuscript. We did not find adding the horizontal bias lines to be helpful so we chose not to add them. However, for our "case study" in Section 6.2 (Figure 10) we added lines showing our estimated uncertainty bounds.

Other minor issues:

-self-sampling for the parked periods is not addressed, but should be

We attempted to minimize self-sampling by instructing the drivers to park into the wind, and have added a statement to Section 3.1 stating this. We recognize that we cannot rule the influence of self-sampling on our results, however, the degree of agreement between the two sites seems to be more influenced by the local traffic emissions, indicating that the effect of self sampling is minimal.  We added text to Section 3.2 stating this.

-did they authors apply the sum of their framework to any of their individual monitors/cars, in order to see if there was any drift/QA issues? This seems like the obvious thing to have done. It at least should be addressed as to why not?

Added a section (Section 6.2) where we show a "real-world" application of this method, including an $NO_2$ sensor with drift versus one without drift, to demonstrate how it could be applied.

-"2.  Overview of Methods" should probably be a bit more precise given the multiple following "Methods" sections, and the fact that some of the text contained in the multiple Results sections is also "Methods." I suggest a change that makee data analysis methods, experimental design, etc.

Renamed Section 2 to "Instrumentation used in this study"

---

## Referee Report (RR1)

Reviewer comments for:

**Mobile air quality monitoring and comparison to fixed monitoring sites for instrument performance assessment**

This review is for the above revised manuscript submitted for publication in Atmospheric Measurement Techniques. The manuscript partially develops and proposes implementation of a new method to evaluate changes in instrument performance in mobile monitoring of air quality. To do that, the authors use high-temporal resolution (O ~ 1s) mobile-monitoring data collected using regulatory-graded instruments from two campaigns conducted in different regions for very different lengths of time for three pollutants, O3, NO2, and NO. The authors then compare stationary referencing of this data during collocations with the regulatory monitor to the referencing of "vehicle-in-motion" concentrations with regulatory monitoring data (based on distance and road type from the regulatory monitor) for one site, and find similar performance evaluations across pollutants for the residential roadtype in their new approach. For the second site, the authors do not conduct stationary referencing and only perform the latter "vehicle-in-motion" referencing to the regulatory monitor to estimate optimal temporal "running windows" to identify instrument issues. They calculate that for a 3 km spatial window, a temporal running window of 40 hours for data would allow detection of a systematic measurement drift or sudden instrument or sensor malfunction over the time scale of 7-9 days. In their revised manuscript, the authors identify and address systematic measurement drift or malfunction by briefly discussing the implementation of this method on another dataset. **I recommend publication of this manuscript following the addressal of the following major and minor comments.**

**Major comments**

**1. Using r2 as a measure of random variability:**

The authors use r2 as a reflection of "the random variability between the mobile and stationary measurements that results from a combination of measurement precision as well as true spatial variability". However, I think what the authors wanted to instead say is that r2 is a measure of "the random variability between the mobile and stationary measurements that results from  measurement precision as well as r2 also captures some true spatial variability". The reason is that systematic (and not random) spatial variability occurs

not just because of road type but several other factors such as wind direction and turbulence regimes which are affected by things such as emission sources and times of day. While I do see the value of r2 in the main manuscript, especially in the context of Figures 3 and 7, it's hard to argue r2 even captures random variability when you are not even sampling the same parcel of air. I suggest that authors explicitly acknowledge the true spatial variability captured by r2 as a systematic and not random variability.

In lines 435-438, the authors say, "The smaller bias, in particular for O3 and OX, could be due to better inter-lab comparability in the California dataset, but aggregating data across multiple sites may also explain a reduction in the systematic bias. For example, if one site has a slightly positive bias and another site has a slightly negative bias (due to monitor siting or random calibration variability), those biases will partially cancel each other out." This sentence made me rethink how clear is r2 a measure of random bias. I am not convinced that r2 in the way it is used (comparison of different air samples at the same time), is a reasonable measure of random spatial variability. **I suggest the authors instead use a cleaner approach to separate systematic and random variability such as the comparison of actual bias and absolute bias.** You could add those comparisons either by adjusting the current panel plots or as supplementary figures, and briefly discussing them (1-2 sentences) wherever using r2 as a measure of random variability is a deficient way of going about it.

2. **Using non-highway as a class versus residential for "mobile to stationary" referencing**

If you look carefully at Figures 3 and 7, it is clear that residential roads are showing stable behavior regardless of distance in terms of mean and median bias. This suggests that they are able to capture a systematic instrument bias that perhaps other road types cannot. The authors identify this in lines 311-312 as "For the Residential road class, the bias between mobile collocation and parked collocation
changes very little as the distance buffer increases for all species." Additionally, in Lines 462-465, I appreciate the authors' effort to highlight the value of data on residential roads. It is then surprising that the authors want to use road type data other than residential to determine detection thresholds of systematic instrument bias/drift/malfunction. I suggest that authors not club residential and other road types, or at the least show in the supplement that just using residential data does not dramatically lower the detection thresholds of instrument malfunction. Otherwise, **that residential roads are a close proxy of stationary collocation is a major finding, is easy to understand, and all figures and discussion should be**

**orientated around that aspect (e.g. Figure 4). This also makes sense in other ways, since health exposure studies naturally sample large sections of residential areas.** This will also address another issue I had with the manuscript which was the lack of results associated with Scenario 1 identified in Section 6. I suggest showing Scenario 1 in the Supplement similar to the analysis showing in Section 6.1, and at least briefly discussing it at the relevant places.

**Minor comments**

1. In addition, the authors say in the responses that "We revised and improved all the maps in the manuscript, including adding shading to indicate parking areas and adding wind roses and scale bars." However, just looking at Figure 1, while I do see parking areas, I neither see wind roses nor scale bars. The authors need to address this issue before publication. Also, please check that you have actually incorporated all aspects that you claim in responses before submitting the final revisions.

2. Figure 3: add number of points in the parked colocations aspect as well. There are no black dots in the bottom subfigures.

3. Figures 5 and 6 do not seem to be particularly useful for the main manuscript. Move them to the supplement.

4. Figure 8 axis labels should clearly state the use of "running medians" in the y-axis.

5. Lines 412-415 "This is in contrast to Sections 3 and 4 where the mean was used to aggregate the
one second data up to one minute or one hour. In general, using the median versus the mean produce similar results for O3, NO2, and OX; however, using the hourly medians versus means significantly reduces the impact of high NO outliers (peaks) on the NO aggregation." Please add supplementary figures showing the difference of mean versus median based figures for Sections 3 and 4. Otherwise, I recommend using consistent underlying central tendency metric across sections as it unnecessary complicates this manuscript for an average reader.

---

## Author Response (AR2)

**Mobile air quality monitoring and comparison to fixed monitoring sites for instrument performance assessment**

Andrew R. Whitehill, Melissa Lunden, Brian LaFranchi, Surender Kaushik, Paul A. Solomon

**Response to Reviewers:**

**Report #1**

Reviewer comments for:

Mobile air quality monitoring and comparison to fixed monitoring sites for instrument performance assessment

This review is for the above revised manuscript submitted for publication in Atmospheric Measurement Techniques. The manuscript partially develops and proposes implementation of a new method to evaluate changes in instrument performance in mobile monitoring of air quality. To do that, the authors use high-temporal resolution (O ~ 1s) mobile-monitoring data collected using regulatory-graded instruments from two campaigns conducted in different regions for very different lengths of time for three pollutants, O3, NO2, and NO. The authors then compare stationary referencing of this data during collocations with the regulatory monitor to the referencing of "vehicle-in-motion" concentrations with regulatory monitoring data (based on distance and road type from the regulatory monitor) for one site, and find similar performance evaluations across pollutants for the residential roadtype in their new approach. For the second site, the authors do not conduct stationary referencing and only perform the latter "vehicle-in-motion" referencing to the regulatory monitor to estimate optimal temporal "running windows" to identify instrument issues. They calculate that for a 3 km spatial window, a temporal running window of 40 hours for data would allow detection of a systematic measurement drift or sudden instrument or sensor malfunction over the time scale of 7-9 days. In their revised manuscript, the authors identify and address systematic measurement drift or malfunction by briefly discussing the implementation of this method on another dataset. I recommend publication of this manuscript following the addressal of the following major and minor comments.

Major comments

1. Using r2 as a measure of random variability:

The authors use r2 as a reflection of "the random variability between the mobile and stationary measurements that results from a combination of measurement precision as well as true spatial variability". However, I think what the authors wanted to instead say is that r2 is a measure of "the random variability between the mobile and stationary measurements that results from a combination of measurement precision as well as r2 also captures some true spatial variability". The reason is that systematic (and not random) spatial variability occurs not just because of road type but several other factors such as wind direction and turbulence regimes which are affected by things such as emission sources and times of day. While I do see the value of r2 in the main manuscript, especially in the context of Figures 3 and 7, it's hard to argue r2 even captures random variability when you are not even sampling the same parcel of air. I suggest that authors explicitly acknowledge the true spatial variability captured by r2 as a systematic and not random variability.

In lines 435-438, the authors say, "The smaller bias, in particular for O3 and OX, could be due to better inter-lab comparability in the California dataset, but aggregating data across multiple sites may also explain a reduction in the systematic bias. For example, if one site has a slightly positive bias and another site has a slightly negative bias (due to monitor siting or random calibration variability), those biases will partially cancel each other out." This sentence made me rethink how clear is r2 a measure of random bias. I am not convinced that r2 in the way it is used (comparison of different air samples at the same time), is a reasonable measure of random spatial variability. I suggest the authors instead use a cleaner approach to separate systematic and random variability such as the comparison of actual bias and absolute bias. You could add those comparisons either by adjusting the current panel plots or as supplementary figures, and briefly discussing them (1-2 sentences) wherever using r2 as a measure of random variability is a deficient way of going about it.

We added text to Section 4.2 (Lines 285 – 295) to clarify what we mean by random spatial variability (which we changed to "spatio-temporal variability") versus persistent spatial biases over time, which we refer to as systematic spatial bias.

2. Using non-highway as a class versus residential for "mobile to stationary" referencing

If you look carefully at Figures 3 and 7, it is clear that residential roads are showing stable behavior regardless of distance in terms of mean and median bias. This suggests that they are able to capture a systematic instrument bias that perhaps other road types cannot. The authors identify this in lines 311-312 as "For the Residential road class, the bias between mobile collocation and parked collocation changes very little as the distance buffer increases for all species." Additionally, in Lines 462-465, I appreciate the authors' effort to highlight the value of data on residential roads. It is then surprising that the authors want to use road type data other than residential to determine detection thresholds of systematic instrument bias/drift/malfunction. I suggest that authors not club residential and other road types, or at the least show in the supplement that just using residential data does not dramatically lower the detection thresholds

of instrument malfunction. Otherwise, that residential roads are a close proxy of stationary collocation is a major finding, is easy to understand, and all figures and discussion should be orientated around that aspect (e.g. Figure 4). This also makes sense in other ways, since health exposure studies naturally sample large sections of residential areas. This will also address another issue I had with the manuscript which was the lack of results associated with Scenario 1 identified in Section 6. I suggest showing Scenario 1 in the Supplement similar to the analysis showing in Section 6.1, and at least briefly discussing it at the relevant places.

Additional analysis on Scenario 1 is outside of the scope of the paper and would require more than just minor revisions to the manuscript. The comparison of two scenarios in Section 6 were intended primarily to introduce the linkage between number of collocation data points and uncertainty in ΔX, contrasting the 500m Residential only scenario with the 3000m Non-Highway scenario. However, the subsequent analysis is focused exclusively on the 3000m Non-Highway data set because it requires a sufficiently large data set to do the sub-sampling. The Non-Highway data set also is expected to result in a more conservative estimate of the uncertainty in ΔX than the Residential only scenario, which is likely more typical across most complex urban environments. We have updated the text in the introduction to Section 6 to reduce the emphasis on introducing two distinct scenarios (1 and 2), which inadvertently raised an expectation of a contrasting analysis of the two for the reader. Instead, we focus the discussion on the differences in the size of the data set between the two examples, and later provide justification for choosing the 3000m Non-Highway scenario for the subsequent analysis.

Minor comments

1. In addition, the authors say in the responses that "We revised and improved all the maps in the manuscript, including adding shading to indicate parking areas and adding wind roses and scale bars." However, just looking at Figure 1, while I do see parking areas, I neither see wind roses nor scale bars. The authors need to address this issue before publication. Also, please check that you have actually incorporated all aspects that you claim in responses before submitting the final revisions.

We remade the maps so that the wind roses and scale bars are clearer (Figures 1, 5, and 6).

2. Figure 3: add number of points in the parked colocations aspect as well. There are no black dots in the bottom subfigures.

Updated figure as suggested (Figure 3).

3. Figures 5 and 6 do not seem to be particularly useful for the main manuscript. Move them to the supplement.

Figure 5 and 6 serve to demonstrate the range of roads driven and the type of driving patterns, which is important to the interpretation of the resulting data. Therefore, we have chosen to keep these images in the main text.

4. Figure 8 axis labels should clearly state the use of "running medians" in the y-axis.

Updated figure as suggested (Figure 8).

5. Lines 412-415 "This is in contrast to Sections 3 and 4 where the mean was used to aggregate the one second data up to one minute or one hour. In general, using the median versus the mean produce similar results for O3, NO2, and OX; however, using the hourly medians versus means significantly reduces the impact of high NO outliers (peaks) on the NO aggregation." Please add supplementary figures showing the difference of mean versus median based figures for Sections 3 and 4. Otherwise, I recommend using consistent underlying central tendency metric across sections as it unnecessary complicates this manuscript for an average reader.

Either measure of central tendency (mean versus median) work for the hourly data aggregation. We wanted to demonstrate examples of using both the mean and the median and believe that either can be used in the resulting analysis with only minimal differences on the results.

**Report #2**

In their manuscript "Mobile air quality monitoring and comparison to fixed monitoring sites for instrument performance assessment", Whitehill and coauthors present analyses from stationary and mobile comparison measurements of ozone, NO, NO2, and Ox (O3 + NO2), measured on-board mobile platforms and in air quality network stations to be used to identify instrument bias during ongoing field measurements with the mobile measurement platforms. This is a strongly revised version of a previous manuscript, in which the data analysis was taken a lot further, compared to the initial submission.

While in the initial version of the manuscript, the authors have mainly based their analysis on linear regression analysis and have presented multiple correlation plots, they picked up my suggestion to show the quality of agreement between air quality network results and their mobile platform measurement results as a function of distance between both measurements for different road types separately. They have taken this further and focus now on average or median bias instead of slope and intercept. In addition, the authors have added a discussion on how well their approach of mobile comparisons could be used for other types of pollutants, depending on their nature and homogeneity of their concentrations in the environment.

The revised version of the manuscript goes well beyond the first version in analysis depth and coherence. In addition, the authors have addressed all relevant points raised in my first review to an acceptable degree. There are several minor points to be worked on, as detailed below. After these points are addressed, I suggest publishing the manuscript in AMT.

Detailed comments:

L24: "… highways showing the most variance." – Shouldn't it be "… highways showing the strongest biases."?

Changed from "most variance" to "largest differences" (Line 24)

L53: "In addition, natural variability …" □ "In addition, natural spatial variability …"

Changed to "natural spatial variability" (Line 55)

L82: "… commonly measured and air quality …" □ "… commonly measured in air quality …"

Changed to "commonly measured at air quality" (Line 84)

L121: "… next-generation air quality instruments": This sounds like very sophisticated air quality instruments. Don't you just mean "low-cost sensors"?

Changed to "low-cost sensors" (Line 123)

L141: The drivers were instructed to park facing into the wind when possible. Was assessed, whether the measurements were affected by the own exhaust under still conditions or when the wind was from the back (which both could also occur during the mobile measurements, e.g., when stopping the vehicle). Were such self-sampling intervals removed from the data sets?

Added: "A visual screening did not reveal any suspected self-sampling periods, so no additional attempts were made to identify or remove such periods." (Line 144 – 145)

L151, Figure 1: From the satellite view and the scale at the bottom of the image, the maximum distance for the CAMP site seems to be rather 35 m and not 85 m.

Added text to caption:

"(Not shown are two periods where the cars parked at the CAMP site just north of the map due to a lack of street parking closer to the site)." (Line 155 – 156)

L162: The QA evaluations showed instrument bias of 3%-6% for O3 and 3%-8% for NO. How do these percentage biases translate into ppbv biases? If they are calculated from the span gas concentrations, 80 and 360 ppbv, this would mean an observed bias of up to 5 ppbv for O3 and up to 29 ppbv for NO. This is larger than the minimum necessary bias for observation as stated in the abstract and in the results section (4 and 8 ppbv). How can this be?

Clarified by adding: "Field bias measurements are based on span checks assuming the linearity of the instrumentation response across the measurement range, which was confirmed in laboratory multi-point calibrations. At the mean observed concentrations during the study (33 ppbv O3 and 53 ppbv NO), the error in span measurements translates to an average bias and precision of 2 ppbv for O3 and an average precision and bias less than 4 ppbv for NO." (Lines 166 – 171)

L206f: The mean or median DeltaX values are claimed to be a more direct assessment of systematic bias than slope and intercept of a linear regression. This is true for an offset in the instrument data. However, a bias due to a change in instrument sensitivity would rather be detected by linear regression than using mean or median DeltaX values.

Clarified that the bias discussed is "offset bias". Also clarified that "This is due to the high sensitivity of OLS regression statistics on outlier points." And "a few extreme outlier points might occur that will skew OLS statistics but have less influence on ΔX statistics". (Lines 215 – 221)

L230: "… differences in concentrations due to the distance between the locations …" - The differences in concentrations are less due to the distance between the locations but rather due to the differences in distance to the sources of emission plumes. Otherwise, it would only result in more scatter, but not in a systematic bias.

Clarified that the difference are due to the "relative proximity of the two instruments to passing emission plumes" (Line 239)

L239: Why does the influence of local traffic emissions reduce the applicability of a linear regression approach but not of a measurement bias approach?

Clarified that "Temporal aggregation can smooth some of the outlier points and make the results more reflective of real measurement differences; however, parametric regression statistics will still be biased by the influence of outlier points." (Lines 247 – 249)

L244-246: The authors state that mobile collocations allow to sample a larger amount of air in the same sampling duration. Mobile measurements sample from the amount of air that passes by the moving vehicle (i.e., depends on difference of velocities of ambient air and vehicle). Stationary measurements sample from the amount of air that passes by the station due to the ambient air movement. On average (i.e., when moving with and against the ambient wind the same amount of time), both are similar in size. In the case that the vehicle moves with the air (in wind direction), the mobile measurement would even probe less volume of air, compared to the stationary measurement.

Added explanation:

"For example, a car traveling 25 meters per second will "sweep" an additional area of 1500 linear meters in one minute compared to the stationary sampling. Thus, regardless of the wind speed, a moving platform will integrate each measurement over a larger area than a stationary platform, making each emission point source have less direct influence on the entire integrated measurement." (Lines 256 – 259)

L257-258: "While travelling on high traffic roads, the cars are more likely to be impacted by direct emission plumes." - This is likely true. Nevertheless, I think that the critical point is the traffic density in the vicinity of the mobile measurement vehicle, not only the road classification. There can also be several cars in front of the mobile measurement on residential roads, e.g., at intersections or traffic lights. This would also strongly affect the measurements of NO and O3. While the road type classification is an easy to perform start, a more sophisticated approach would be desirable for the future.

We agree, but this is beyond the scope of the present study. Added the following text;

"Although more sophisticated methods are possible to identify and remove high traffic roads, OSM road classifications are a general proxy that can be applied algorithmically over a large portion of the Earth. In contrast, local traffic count data are more sporadic and not always available or easily accessible for the region of interest." (Lines 274 – 276)

L271: "… bias values can reflect …" ☐ "… bias values from in-motion collocations can reflect …"

Clarified that "bias values from both stationary and in-motion collocations can reflect…" (Line 286)

L272: I would remove the "random" in "… generally reflect the random variability …" because non-random variability, e.g., due to persistent spatial differences, would also be reflected in r2.

We clarified Section 4.2 to better define what we mean by "random" variability (Section 4.2)

L288: "the relationship" seems to be not the right expression since there is no interaction between the stationary and the mobile collocation measurements. Better something like "in comparison to".

Replace "the relationship" with "the similarity"

L292: The larger magnitude of bias for the Highway road class, compared to the other road classes, does not indicate larger spatial variability (this would be shown by larger r2 values) but larger (or smaller in case of O3) average or median concentrations.

Changed to "… indicating a larger influence of direct emission plumes increasing the variability in concentrations measured on Highways…" (Line 315)

L305, Figure 3: Please use consistently in the text and in the Figures either r2 or R2.

Corrected to use $r^2$ consistently throughout the manuscript.

L306: What does "maximum distance" mean in this context: are the median and mean bias values taken from all measurements up to the respective distance buffer shown, i.e., for larger distance buffers also the measurements from closer to the fixed site are included; or are only the increments in distance buffer used as basis for the respective data? The former provides only very indirect information on how the measured bias depends on distance to the fixed site. It should be more clearly stated, which data are included in the individual data points.

Added clarification (L308-311):

"Note here that for each distance D we include all datapoints within a distance of D from the stationary site, so we are not explicitly showing how bias varies with distance from the stationary monitors."

This approach is not very helpful in the analysis of how strongly spatial variability affects the comparison measurement, because it is strongly affected by the number of data points in the individual distance buffer increments. E.g., for residential roads there seem to be barely any data points available beyond a distance of 2 km. This results in barely any change in bias values beyond this distance, which is not a result of spatial homogeneity (as it seems) but a result of a lack in data points in this distance range. Normalizing the contribution from individual distance buffer increments by the number of data points within the respective increments would provide a more direct information on the influence of distance on the comparability of the measurements and would therefore allow to apply the results also to other environments, where the distribution of various road types might be different.

Because our method specifically looks at all datapoints within a set distance (maximum distance) from the site while averaging we felt it was more appropriate to show the data as we have in the manuscript.

L326-331: In order to judge whether the observed bias values are significant and whether they would allow a reasonable identification of measurement bias, it would be interesting to also see the average absolute concentration values and their variability during the measurements. This is shown in Figure S4 and S5, but this information would be essential in the main text as well.

This information is shown in Figures S6 – S9 in the supplement and referenced in the main text.

Furthermore, according to the time series, e.g., for NO it looks like that rather than average or median values the minimum or better something like the 5% percentile would represent the measurements not affected by local plumes (i.e., those of the stationary sites) much better.

It is possible that a 5$^{th}$ percentile might work well for some species, such as NO, but is unlikely to work for others, such as $O_3$. Exploration of alternative metrics is beyond the scope of this manuscript.

L426: "… random spatial biases." □ "… random spatial variability."

Replace "random spatial biases" with "random spatio-temporal variability"

L433: I agree that the higher r2 for NO2 could be due to the larger dataset used, however, it also may be due to the fact that only 1-hour averages were used there, where short concentration peaks are largely averaged out.

The comparisons here are made to the 1-hour averaged data in Figure 4, which uses the same 1-hour averaging period. Clarified that the comparisons are made to the "equivalent hour-averaged results for the Denver study". (Line 458)

L439f: This statement again shows that without taking the number of data points per road type or distance increment into account, the results reflect to a certain degree the distribution of road types and not necessarily the actual spatial variability of pollutant concentrations.

Added the following sentence:

"Variances in traffic patterns, road type distributions, and other factors could also influence differences in biases in different geographic regions or using different driving patterns, so the range of biases must be measured for each individual study region and study design." (Lines 464 – 467)

L458: Indeed, it seems advantageous to remove Highway road segments from the data. However, it also would probably be advantageous to remove all data points which are from short-time peaks (i.e., from plumes of nearby sources) – also from the data of the other road types.

Added the following clarification:

"Although more complex peak-removal algorithms can achieve similar goals, they add complexity without necessarily improving the data and add additional arbitrary bias (e.g., "cherry-picking") to the resulting comparisons." (Lines 489 – 491)

L491-496: It is not really clear to me how the upper and lower traces were calculated. Are these the lowest and highest 1-hour medians within the running median? Or is this the minimum and maximum of the running medians (over different window sizes) of the 1-hour medians? It would be desirable, if this is explained a bit more clearly.

Added the following sentence to clarify:

"For each running median window size N, the upper and lower traces reflect the maximum and minimum of the set of running N-hour medians from this dataset." (Lines 525 – 526)

L519-525: So, this means that under typical conditions, the response time would be likely several weeks, correct? In this case, wouldn't it be easier to have a quick calibration check every couple of weeks where a calibration gas mixture is probed for a couple of minutes by the instrument setup, compared to the ongoing analysis of mobile collocations with their higher bias uncertainty

Added clarification to introduction:

"While the use of fleets facilitates scaling of mobile monitoring to large geographic scales, such as multiple counties in an urban area or multiple cities across a large state, coordinating vehicles and drivers to across these geographies makes route laboratory-based calibrations costly, time consuming, and impractical." (Lines 38 – 40)

L567: Why are here averages of data instead of medians (as in the rest of the manuscript) used?

We recognize that using means instead of medians could create confusion. However, we wanted to demonstrate that running medians of hourly means versus hourly medians both work and produce the desired outcome.

L577; Figure 10: Why does the 40-hour running median start (and end) at the same time as the data points start (and end)? Shouldn't there be a lag in the start with the running median starting after 40 hours only (as in the previous figure)?

Figure was updated based on this suggestion. (Figure 10)

L589: How does the analysis has implications for the spatial heterogeneity? I guess the latter one is unaffected by the analysis.

Replaced "has implications for" with "provides information about"

L608: Removing highway-related data does not reduce the spatial heterogeneity of pollutant concentrations, it reduces the influence of local emission plumes onto the measurement data.

Replaced "spatial heterogeneity" with "influence of local emission plumes" (Line 643)

L633: Not "the influence of highways" needs to be removed, but the influence of emission plumes on the measurements – which are more frequent on highways, compared to other road types.

Clarified that the purpose is to "reduce the influence of emission plumes, which are most abundant on highways" (Lines 668 – 669)